# Cortical overgrowth in a preclinical forebrain organoid model of *CNTNAP2*-associated autism spectrum disorder

Job O. de Jong[1,2], Ceyda Llapashtica[1,2], Matthieu Genestine[1,2], Kevin Strauss[3], Frank Provenzano[4], Yan Sun[1], Huixiang Zhu[1,2], Giuseppe P. Cortese[1,2], Francesco Brundu [5], Karlla W. Brigatti[3], Barbara Corneo [6], Bianca Migliori[7], Raju Tomer [8], Steven A. Kushner [1,9], Christoph Kellendonk[1,2,10], Jonathan A. Javitch [1,2,10], Bin Xu[1,2✉] & Sander Markx[1,2✉]

We utilized forebrain organoids generated from induced pluripotent stem cells of patients with a syndromic form of Autism Spectrum Disorder (ASD) with a homozygous protein-truncating mutation in *CNTNAP2*, to study its effects on embryonic cortical development. Patients with this mutation present with clinical characteristics of brain overgrowth. Patient-derived forebrain organoids displayed an increase in volume and total cell number that is driven by increased neural progenitor proliferation. Single-cell RNA sequencing revealed PFC-excitatory neurons to be the key cell types expressing *CNTNAP2*. Gene ontology analysis of differentially expressed genes (DEgenes) corroborates aberrant cellular proliferation. More-over, the DEgenes are enriched for ASD-associated genes. The cell-type-specific signature genes of the *CNTNAP2*-expressing neurons are associated with clinical phenotypes previously described in patients. The organoid overgrowth phenotypes were largely rescued after correction of the mutation using CRISPR-Cas9. This *CNTNAP2*-organoid model provides opportunity for further mechanistic inquiry and development of new therapeutic strategies for ASD.

[1] Department of Psychiatry, Vagelos College of Physicians & Surgeons, Columbia University, New York, NY, USA. [2] Division of Molecular Therapeutics, New York State Psychiatric Institute, New York, NY, USA. [3] Clinic for Special Children, Strasburg, PA, USA. [4] Taub Institute for Research on Alzheimer's Disease and the Aging Brain, Department of Neurology, Columbia University, New York, NY, USA. [5] Department of Systems Biology, Columbia University, New York, NY, USA. [6] Stem Cell Core Facility, Columbia University, New York, NY, USA. [7] Tech4Health and Neuroscience Institutes, NYU Langone Health, New York, NY, USA. [8] Department of Biological Sciences, Columbia University, New York, NY, USA. [9] Department of Psychiatry, Erasmus MC University Medical Center, Rotterdam, The Netherlands. [10] Department of Pharmacology, Vagelos College of Physicians & Surgeons, Columbia University, New York, NY, USA. ✉email: bx2105@cumc.columbia.edu; sm2643@cumc.columbia.edu

Autism spectrum disorder (ASD) and related neurodevelopmental disorders represent a major public health burden[1]. To better understand these conditions, mechanistic insight into their development is needed. Rodent models with modified genes associated with ASD have facilitated this effort, but translating promising preclinical findings to the clinic has remained challenging[2]. This is at least partly due to differences in the biology between species[3–5] and underscores the need for elucidating human-specific disease mechanisms for psychiatric disorders. Forebrain organoids generated from human induced pluripotent stem cells (hiPSCs) derived from subjects with well-characterized brain disorders are a promising method to study human-specific neurobiology. Indeed, brain organoid modeling of ASD and related neurodevelopmental disorders may help reveal critical disease phenotypes and associated disease mechanisms that may not be recapitulated in animal models[6–9].

A homozygous loss-of-function (LoF) mutation, c.3709DelG, in contactin-associated-protein-like 2 (CNTNAP2) causes a rare and severe neurodevelopmental syndrome characterized by ASD, intellectual disability, early-onset epilepsy and an increase in head circumference – collectively termed Cortical Dysplasia Focal Epilepsy (CDFE) syndrome[10]. The inherited mutation and associated clinical syndrome were first described in a family from the genetically isolated Old-Older Amish population[10]. Additional cases carrying different homozygous LoF mutations were subsequently discovered in subjects outside the Amish founder population, all presenting with a similar constellation of symptoms[11,12]. Thus far, no homozygous LoF mutations have been described in healthy control individuals, indicating that these are associated with a severe and a highly penetrant phenotype.

CNTNAP2 encodes a cell adhesion molecule that is structurally related to the neurexins, but has shown to be functionally distinct, fulfilling roles in both peripheral and central nervous system development and physiology[12]. Known functions of CNTNAP2 have been delineated in a number of Cntnap2 knock-out and knock-down rodent models, and include critical roles in potassium channel clustering in myelinated axons[13,14] as well as dendritic arborization[14], synaptic development and function[14,15] and neuronal migration[16]. Interestingly, Cntnap2 null mice display several behavioral phenotypes reminiscent of the neurodevelopmental syndrome associated with homozygous LoF mutations, including deficits in social interaction and cognition, an increase in repetitive-stereotyped behaviors, and a lowered seizure threshold[16].

Little is known about the function of CNTNAP2 in the human brain. Human expression data shows high abundance in different cortical areas during embryonic development[17]. Histopathology and MRI findings from patients from the Old-Order Amish population diagnosed with CDFE-syndrome show cortical dysplasia, cortical gray matter thickening, ectopic neurons in the white matter tracts, and increased hippocampal cell number[10]. Together, these findings imply an early embryonic origin of pathophysiology. As embryonic development is faithfully recapitulated in brain organoid modeling systems as shown by transcriptome analyses[18,19], we generated forebrain organoids to study how disruption of CNTNAP2 affects cortical embryonic development.

To this end, we generated hiPSC lines from subjects from the Old-Order Amish founder population, all of whom had been diagnosed with CDFE-syndrome and were genetically confirmed to carry the homozygous c.3709DelG mutation in CNTNAP2. Using these lines, we generated forebrain organoids and compared these to organoids generated from healthy controls from the same population to examine the effects of the mutation on cortical development. We show an increase in volume of patient-derived organoids compared to controls. This enlargement of organoid volume results from increased proliferation of neural progenitor cells (NPCs) and other proliferating cells, which leads to an increase in total cell number. Single-cell RNA sequencing (scRNAseq) revealed PFC-excitatory neurons to be the key cell types expressing CNTNAP2. Differential gene expression analyses at both the bulk- and single-cell level corroborate changes in biological processes including cell proliferation and neuronal differentiation. Differentially expressed genes were enriched for ASD-associated genes and Weighted Gene Co-Expression Networks (WGCNAs). The cell type-specific signature genes in CNTNAP2-expressing neural cell types are associated with multiple clinical disease phenotypes that have previously described in patients with CNTNAP2 LoF mutations. Together, our work provides a critical step towards understanding the role of CNTNAP2 in human brain development. These findings provide possible mechanisms underlying the CDFE syndrome, including the increased head circumference and cortical gray matter thickening, epilepsy and impairment in language development[10]. Future studies should focus on further elucidating these disease mechanisms which can, in turn, lead to the development of novel treatments of CNTNAP2-associated ASD and other genetic subtypes of ASD.

## Results

**Clinical description of patients carrying c.3709DelG mutation in CNTNAP2.** Patients carrying the c.3709DelG mutation in CNTNAP2 were reported to have increased head circumference, epileptic seizures, language regression, intellectual disability, and ASD[10]. Because the sample size of the initial cohort in which the increased head circumference was described was small[10], we expanded this assessment by measuring the head circumferences of an additional 17 male and 20 female pediatric patients from the Old-Order Amish founder population all carrying the inherited homozygous c.3709DelG mutation in CNTNAP2 and all presenting with the same syndromic ASD. We compared these data to head circumference reference data published by the WHO[20]. This analysis revealed an increased head circumference in both male and female patients compared to the WHO mean (weighted $z$-score = 2.93, $p < 0.003$ (Fig. S1A). To expand upon the initial description of increased gray matter thickness[10], we subsequently analyzed brain MRIs from six patients carrying the c.3709DelG mutation in CNTNAP2 and compared these to age-matched clinical templates. We found an increase in total gray matter (GM) volume relative to total brain volume (TBV) in patients compared to controls (Patients: GM/TBV = 0.73; Controls: GM/TBV = 0.54; $t$-test, $p = 0.004$, $n = 6$ patients, 4 control templates (Fig. S1B and Supplementary Data 3).

**hiPSC and forebrain organoid generation and characterization.** It has been previously shown that CNTNAP2 expression is highest in frontal- and anterior cortical regions in the human embryonic brain[17]. We therefore focused on investigating the role of CNTNAP2 on forebrain development by generating forebrain organoids. We generated hiPSC lines from monocytes of three human female patients (ages 10, 12, and 17 years) carrying the homozygous c.3709DelG mutation in CNTNAP2 (Fig. 1a) and three female healthy control subjects from the same population. All three patients were assessed by a pediatrician (KS) and were diagnosed with epilepsy and a syndromic form of ASD, characterized by cognitive-, behavioral-, and language regression and delayed developmental milestones starting around age 1.

To confirm stemness in hiPSC lines used for organoid generation we performed RT-PCR for markers NANOG and OCT4 (Fig. S1C); we performed G-band karyotype analysis to

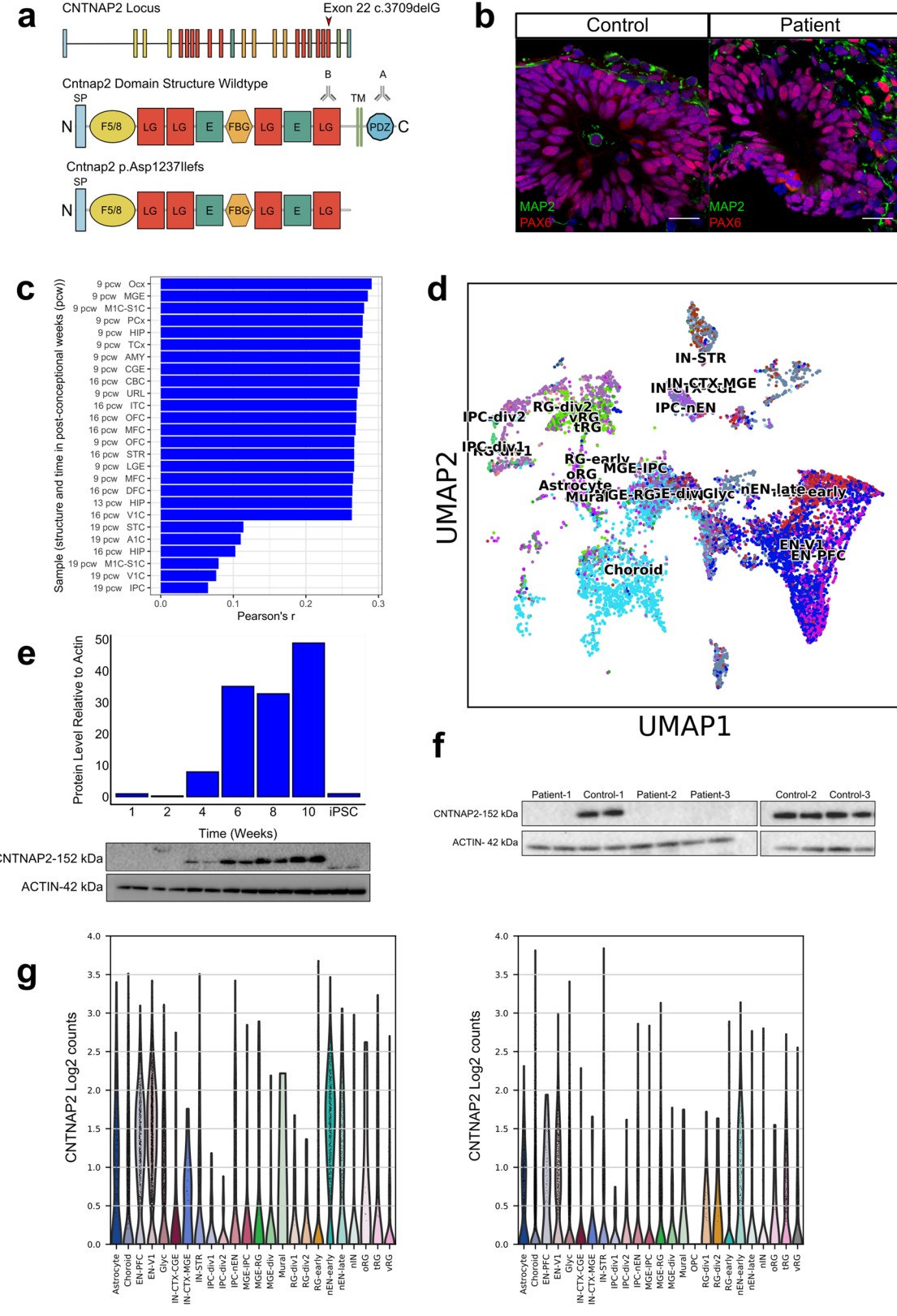

ensure the absence of chromosomal abnormalities in all patient- and control-derived cell lines (Fig. S1D). We confirmed the genotypes of patient- and control-derived hiPSCs using Sanger sequencing (Fig. S1E).

We deployed three approaches to determine whether our forebrain organoid model recapitulates key characteristics of early telencephalic brain development. First, after 4 weeks in culture, immunostaining of both patient- and control-derived organoids showed ventricle-like structures composed of NPCs expressing forebrain-specific transcription factor PAX6 (Fig. 1b). Dorsal forebrain identity was further confirmed by immunostaining for transcription factor FOXG1 (Fig. S1F). Organoids from both

**Fig. 1 Forebrain organoid characterization and *CNTNAP2* expression. a** *CNTNAP2* locus and protein domain structures of wildtype and mutant protein. The 24 exons of *CNTNAP2* are color-coded based on domain structures of the full-length protein. Deletion of a guanine base at position 3709 in exon 22 (indicated with red arrow) of the gene results in shift in open reading frame and a stop codon at amino acid position 1253. The truncated variant of the protein (p.Asp1237Ilefs) is terminated right before the C-terminal transmembrane domain. Antibody A (intracellular) is a polyclonal antibody directed against the C-terminal end of the protein. Antibody B (extracellular) is a monoclonal antibody directed against the Laminin-G domain upstream of the transmembrane domain (also see Supplementary Data 1 for antibodies). SP signal peptide, F5/8 discoidin/neuropilin homology domain, LG Laminin G domain, E EGF-Like Domain, FBG Fibrinogen-Like Region, PDZ PDZ-interaction domain. **b** Both control-derived and patient-derived organoids show PAX6-expressing dorsal forebrain neural progenitor cells in ventricular zone-like regions after 4 weeks in culture. Neuronal dendrosomatic marker MAP2-expressing cells are preferentially located in the outer zones of the cortical organoids. All control- and patient-derived organoids displayed similar expression patterns for these markers. Scale bar represents 50 μm. **c** Bar plot displaying Pearson's *r* correlations of the bulk RNAseq expression profile of 8-week-old wild-type organoids compared to all publicly available expression profiles of multiple brain structures across multiple time points from BrainSpan. Displayed are the top-20 highest correlating structures and the 6 lowest. The highest correlations are for multiple forebrain structures at 9 post-conceptional weeks, indicating that the organoids have the highest resemblance to these structures at the transcriptional level. (OCx occipital neocortex, MGE medial ganglionic eminence, M1C-S1C primary motor-sensory cortex, PCx parietal neocortex, HIP hippocampus, TCx temporal neocortex, AMY amygdaloid complex, CGE caudal ganglionic eminence, CBC cerebellar cortex, URL upper (rostral) rhombic lip, ITC inferolateral temporal cortex (area TEv, area 20), OFC orbital frontal cortex, MFC anterior (rostral) cingulate (medial prefrontal) cortex, STR striatum, LGE lateral ganglionic eminence, MFC anterior (rostral) cingulate (medial prefrontal) cortex, DFC dorsolateral prefrontal cortex, V1C primary visual cortex (striate cortex, area V1/17), STC posterior (caudal) superior temporal cortex (area TAc), A1C primary auditory cortex (core), IPC posteroventral (inferior) parietal cortex. **d** UMAP plot of combined control samples (*n* = 2 samples; 11,328 cells) of 13-week-old brain organoids displaying the distribution of various cell lineages annotated according to Nowakowski et al. (ref. [22]). **e** Western blot analysis showing increasing CNTNAP2 protein levels over time relative to actin in a control-derived organoid line, roughly corresponding to the developmental pattern of CNTNAP2 expression pattern in human embryonic brain. Each sample contains two lanes with technical replicates; the bar graph represents mean quantification from these two lanes. Source data are provided as a Source Data file. **f** Western blot analysis showing presence of full-length CNTNAP2 protein levels in control-derived and absence in patient-derived forebrain organoids using an antibody directed against the C-terminal end of the protein (Antibody A in panel **a**). Each sample contains two lanes with technical replicates. Source data are provided as a Source Data file. **g** Violin plot displaying *CNTNAP2* mRNA log2 counts for cell types in control organoids (left panel) and case organoids (right panel). *CNTNAP2* is most highly expressed in EN-PFC, EN-V1, nEN-early, and nEN-late in control organoids (EN-PFC Early Born Deep Layer/subplate Excitatory Neuron PFC, EN-V1 Early Born Deep Layer/subplate Excitatory Neuron V1, Glyc glycolysis, IN-CTX-CGE1 CGE/LGE-derived inhibitory neurons, IN-STR striatal interneurons, IPC-div1 Dividing Intermediate Progenitor Cells RG-like, IPC-div2 Intermediate Progenitor Cells RG-like, IPC-nEN Intermediate Progenitor Cells EN-like, MGE-IPC MGE Progenitors, MGE-RG MGE Radial Glia 1, MGE-div dividing MGE Progenitors, RG-div1 Dividing Radial Glia (G2/M-phase), RG-div2 Dividing Radial Glia (S-phase), RG-Early early radial glia, U1 unknown1, nEN-early Newborn Excitatory Neuron - early born, nEN-late Newborn Excitatory Neuron - late born, nIN MGE newborn neurons, oRG outer radial gla, tRG truncated radial glia, vRG ventricular radial glia) annotated according to Nowakowski et al. (ref. [22]).

genotype groups demonstrate a clear pattern of MAP2-expressing neurons positioned around the ventricle-like structures and preferentially located in the outer layers of the organoids, indicative of initial NPC differentiation and subsequent migration towards the outer cortex-like region (Fig. 1b). At 13 weeks in vitro, organoids displayed expression of early born cortical layer marker TBR1 and CTIP2 as shown using immunostaining (Fig. S1G).

Second, at the transcriptional level, we performed bulk RNA sequencing to reveal that at 8 weeks in culture, the organoids display the highest correlations with multiple forebrain structures from 9-week-old human fetal brain tissue, when compared to transcriptional profiles from samples from all brain structures and timepoints from the Brainspan transcriptome database[21] (Fig. 1c). Third, we evaluated whether the cell type composition of the organoids resembles that of early human fetal prefrontal brain, by performing scRNAseq on 13-week-old organoids. Unsupervised Leiden clustering revealed 27 distinct clusters that were consistently present in all case- and control-derived organoids (Fig. S1H and Supplementary Data 4). We annotated these cell populations using PFC cell-type gene expression signatures previously identified by scRNAseq of human embryonic PFC tissue (average age: 16.3 post conceptional weeks (pcw))[22]. These analyses demonstrate that the forebrain organoids contain a range of embryonic prefrontal cortical cell types (Fig. 1d; Fig. S1I, J; and Supplementary Data 5).

**CNTNAP2 expression in forebrain organoids**. To determine the *CNTNAP2* expression pattern in control-derived organoids, we performed western blotting, which revealed that protein expression starts around 4 weeks in culture and then increases over time

(Fig. 1e). Western blotting for CNTNAP2 utilizing an antibody (Antibody A, intracellular, Fig. 1a) directed against the c-terminus that is deleted in the c.3079delG mutation showed that full-length CNTNAP2 protein was not detected in patient-derived organoids while it was indeed detected in control-derived organoids (Fig. 1f). We then used an antibody directed against the laminin G domain of the protein located upstream of the transmembrane domain (Antibody B Fig. 1a), which is predicted to be present in the truncated CNTNAP2 protein. Notably, while immunoblotting with antibody B revealed *CNTNAP2* expression in the control samples, no protein was detected in patient-derived organoids suggesting that the truncated *CNTNAP2* mRNA and/or protein are unstable (Fig. S1K). Analyzing the mRNA sequence reads for *CNTNAP2* scRNAseq confirms that the mRNA is indeed present in lower numbers, with case organoids showing a lower read-coverage at the 3′ end than controls (Fig. S1L). To compare these expression data in organoids to expression in the intact brain, we analyzed data from the publicly available gene-expression database BrainSpan, which characterizes the embryonic spatiotemporal expression pattern of the developing human brain. These data show the highest *CNTNAP2* expression levels in the first trimester of embryonic development in multiple cortical areas, including orbitofrontal, prefrontal, and temporal cortices (Fig. S1M). Other areas of high gene expression during embryonic development include the striatum and cerebellar cortex (Fig. S1M). The BrainSpan data are in line with previous expression studies of *CNTNAP2* in the human embryonic brain[17,23]. Analysis of scRNAseq data indicates that at 13 weeks in vitro, in control organoids, *CNTNAP2* is most highly expressed in different types of PFC-excitatory neurons as annotated previously:[22] 'Early Born Deep Layer/subplate Excitatory Neuron

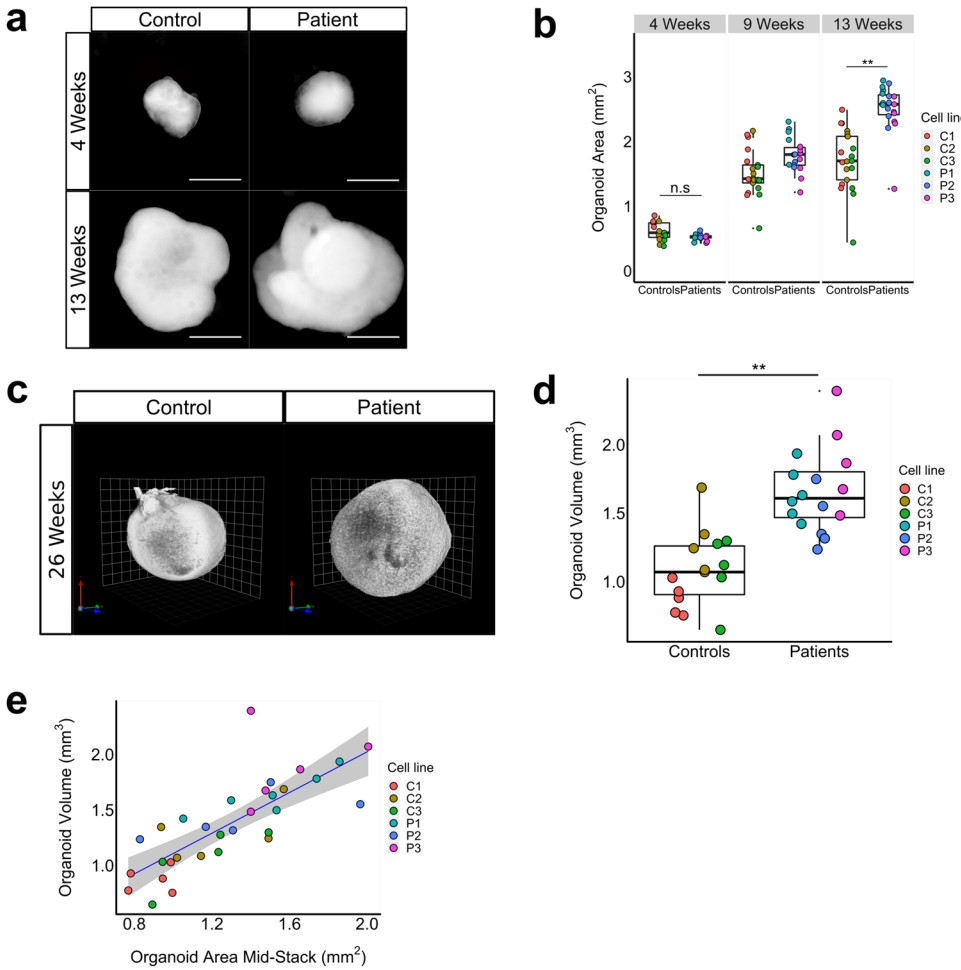

**Fig. 2 Patient-derived forebrain organoids display an increase in volume. a, b** Representative image of bright field images of patient- and control-derived organoids over time (**a**) and quantifications based on 2D bright field image measurements (**b**), showing an increase in surface area at 9 weeks and a 1.5-fold increase in area of patient-derived versus control-derived organoids at week 13 (Likelihood Ratio Test (LRT), $p = 0.002$, $\chi2(1) = 9.42$, $n = 3$ cell-lines/genotype, C1: $n = 7$ organoids (orgs), C2: $n = 7$ orgs, C3: $n = 7$ orgs, P1: $n = 6$ orgs, P2: $n = 6$ orgs; P3: $n = 6$ orgs), while no difference in size was observed at week 4. Scale bar represents 500 µm. **c, d** Representative image of light-sheet microscopy images of control- and patient-derived cortical organoids at 26 weeks in vitro (**c**) and volumetric quantification of light-sheet images showing a 1.5-fold volumetric increase for patient-derived cortical organoids compared to controls (**d**); (LRT, $p = 0.005$, $\chi2(1) = 7.85$, $n = 3$ cell-lines/genotype, C1,C2,C3,P2,P3: $n = 5$ orgs, P1: $n = 6$ orgs). **e** Scatterplot showing the relationship between the area of a middle z-stack of a light-sheet image of a forebrain organoid and the volume at 26 weeks as measured by the 3D rendering of the light-sheet image. We constructed a linear model of volume as a function of the area of the middle z-stack (blue line) ($F(1,29) = 44.59$, $R^2 = 0.606$, $p = 2.529e-07$). The shaded gray area represents the constructed 95% confidence interval around the mean. This result indicates that 2D bright field microscope images can be used to reliably estimate organoid volume. Boxplots display median, first and third quartiles, and whiskers showing the largest and smallest values no further than 1.5 times the inter quartile range from first and third quartile, respectively. *$p < 0.05$; **$p < 0.01$; ***$p < 0.001$.

PFC' (EN-PFC), 'Early Born Deep Layer/subplate Excitatory Neuron V1' (EN-V1) and 'New born excitatory neurons' (nEN) (hereafter referred to as '*CNTNAP2*-expressing cell types'). In patient-derived organoids, *CNTNAP2*-expression is markedly lower in these cell types (Fig. 1g). This is in line with cell type expression pattern in human PFC cell types, with *CNTNAP2* being expressed most highly in excitatory and inhibitory neuronal cell types (Fig. S1n).

These results indicate that the *CNTNAP2* expression pattern in control organoids are in accordance with in vivo gene expression and thus indicate that forebrain organoids serve as a well suited preclinical neuronal model system to study the effects of this homozygous LoF-mutation on early human cortical development.

**Patient-derived forebrain organoids display an increase in volume.** To investigate whether the brain overgrowth characteristic of these patients could be recapitulated in our model system,

we set out to assess the size of patient- and control-derived organoids over time by measuring the projected surface area of the organoid using 2D bright field microscopy images (Fig. 2a). After 4 weeks in culture, case- and control organoids that were generated from an equal number of hiPSCs displayed no difference in surface area. At 9 weeks, case-derived organoids demonstrates an increase in surface area relative to control-derived organoids, and at 13 weeks in vitro, patient-derived organoids showed a 1.5-fold increase in 2D projected surface area compared to control-derived organoids (LRT, $p = 0.002$, $n = 6-7$ organoids/line) (Fig. 2b).

As the organoids are not spherical but irregularly shaped, we used light-sheet microscopy to more precisely quantify organoid volume (Fig. 2c). We found a 1.5-fold volumetric increase for patient-derived organoids compared to controls (LRT, $p = 0.005$) (Fig. 2d). We then utilized light-sheet imaging to investigate whether the total surface area of the middle z-section could be

used as an reliable estimate for organoid volume using a linear model, which was indeed the case (Linear model, $R^2 = 0.606$, $p < 0.001$) (Fig. 2e). We therefore proceeded using 2D brightfield microscopy images for quantifying organoid volume in all following analyses.

**Increased organoid volume is driven by an increased total cell number caused by increased proliferation of dividing cells.** To establish the factors that contribute to the volumetric increase of the patient-derived organoids, we utilized isotropic fractionation[24] to estimate the total cell number in the organoids. We found a 2.5-fold increase in total cell number after 6 weeks in culture for patient-derived organoids compared to controls (LRT, $p < 0.001$, $n = 4$ organoids/line) (Fig. 3a). As expected, organoid surface area and total cell number correlated positively with each other at both 6- and 13 weeks in vitro (Fig. 3b and Fig. S2A). To determine whether cell size also contributes to the volumetric increase of the organoids, we measured the neuronal soma of AAV-Synapsin-GFP transduced organoids, and found no difference in neuronal soma volume between genotype groups (LRT, $p = 0.69$) (Fig. S2B). We then estimated the cell cycle length by co-labeling with BrdU and Ki67, and found that proliferating cells have a shorter cell cycle (Average cell cycle time (Tc) = 54 h) for patient-derived organoids compared to controls (Tc = 83 h) (LRT, $p = 0.02$, 1.54-fold change = 1.54, $n = 3$ organoids/line) (Fig. 3c, d). Co-labeling of BrdU with PAX6 confirmed that most proliferation took place in the PAX6 + NPCs (Fig. S2C) in organoids from both genotype groups (LRT, $p = 0.46$, $n = 3$ organoids/genotype). Next, we quantified the size of the progenitor pool by calculating the absolute numbers of PAX6 + cells at 6 weeks in vitro. We found a 2.6-fold increase in absolute numbers of NPCs in the patient-derived organoids compared to controls (LRT, $p < 0.001$, $n = 4$ organoids/line) (Fig. 3e–g). These data show that increased proliferation activity of NPCs and other proliferating cells leads to an increase in the progenitor pool, which is a major contributor to the volumetric increase observed in patient-derived organoids. At 13 weeks in vitro, the scRNAseq data reveal a decrease in the unsupervised clusters that represent *CNTNAP2*-expressing neuronal cell types (EN-PFC, EN-V1, nEN) in patient-derived organoids (Fig. 3h and Fig. S2D, E). A possible explanation could be a deficit in neuronal differentiation towards these cellular fates; alternatively, the increase in glycolysis cells ("Glyc") in patient-organoids could imply an increase in cellular stress in these cell types (Fig. S2E and Fig. 1g). This could be cell type specific due to the mutation in *CNTNAP2* and/or a property intrinsic to the brain organoid culture as described previously[25].

**Gene-ontology analysis based on bulk RNAseq and scRNAseq data corroborates abnormal cellular proliferation and neurogenesis processes.** To evaluate the transcriptome changes associated with the phenotypes described above, we conducted bulk RNA sequencing (RNAseq) on 8-week-old organoids. Principal component analysis (PCA) of the RNAseq data shows a clear separation in the transcriptional pattern between the two genotype groups, with 47% of the variance explained by principal component 1 (Fig. 4a). Differential gene expression analysis identified a total of 339 differentially expressed genes (false discovery rate adjusted *p*-value < 0.05), including 89 upregulated- and 250 downregulated genes in patient-derived organoids compared to controls (Fig. 4b) (Supplementary Data 6). To investigate whether these differentially expressed genes have known roles in biological processes related to the observed *CNTNAP2*-associated disease phenotypes, we performed Gene Ontology (GO) analysis[26] and found a statistically significant

enrichment for genes involved in various biological processes, including cell proliferation and neurogenesis (Fig. 4b, S3A and S3B, and Supplementary Data 7 and 8). We then performed GO analysis on the cell type-specific signature genes for each of the 3 *CNTNAP2*-expressing cell types (EN-PFC, EN-V1, and nEN-early) in the control organoids from the scRNAseq data at 13 weeks. These genes were enriched for involvement in biological processes including neuronal differentiation (Fig. S3C–E).

**Differentially expressed genes show enrichment for ASD-associated genes and ASD Weighted Gene Co-expression Networks.** As ASD has been described as one of the core features of the neurodevelopmental syndrome associated with the homozygous c.3709DelG *CNTNAP2* mutation in the Old-Order Amish patients[10], we set out to investigate whether affected genes are enriched for genes with an established association with ASD. An enrichment could suggest that different genetic predispositions could lead to autism through effects on overlapping sets of genes. To do so, we compared the differentially expressed genes from the bulk-sequencing experiment to the ASD-associated genes curated by SFARI Gene[27]. Of the 980 genes in SFARI categories 1–4 and the category titled 'syndromic', 19 were differentially expressed between patient- and control-derived organoids (Fig. 4c and Supplementary Data 9). To test whether this enrichment reached the threshold for statistical significance, we performed permutation analyses ($n = 10,000$). In each iteration, we generated a random gene list of equal length to the differentially expressed genes and found that this was indeed the case ($z = 2.42$, $p = 0.016$) (Fig. 4d). We then performed pseudo-bulk differential expression analysis on the scRNAseq data, calculating differential gene expression in the three *CNTNAP2*-expressing cell types (EN-PFC, EN-V1, nEN-early) (Supplementary Data 10). The enrichment for the same SFARI genes was more pronounced among differentially expressed genes in these cell types, where 204 out of 2800 differentially expressed genes were genes present in the SFARI database ($z = 13.27$, $p = 3.42e-40$) (Fig. 3e, f and Supplementary Data 11) implying that transcriptional dysregulation of ASD-associated genes occurs in *CNTNAP2*-expressing cell types.

We subsequently used the same approach to test whether the differentially expressed genes discovered by bulk sequencing were also enriched in three independently created Weighted Gene Co-Expression Networks (WGCNAs) for ASD[28–30]. These WGCNAs were constructed based on the correlation of gene-expression levels between genes, and led to the generation of gene co-expression modules that are associated with specific biological or disease processes[31]. We found a statistically significant enrichment for all three WGCNAs (Fig. S3F–H), indicating an overlap in transcriptional signature between the genes differentially expressed between patient and control-derived organoids and other forms of ASD. We then compared the differentially expressed genes to gene regulatory network data publicly available for other neuropsychiatric conditions, such as Alzheimer's disease (AD)[32] and Bipolar Disorder (BD)[33], and found no statistically significant enrichment for these WGCNAs (Fig. S3I–J). This suggests that the enrichment of differentially expressed genes in WGCNAs may be specific to ASD and related neurodevelopmental disorders. The enrichment for ASD WGCNAs was more pronounced as well for the DE genes in *CNTNAP2*-expressing cell types (Fig. S3K–M). Disease phenotype GO analysis[34] on the cell type-specific signature genes for each of the three *CNTNAP2*-expressing cell types are associated with ASD, epileptic encephalopathy, Pitt-Hopkins syndrome, and hypoplasia of the corpus callosum (Fig. S3N–P). The first three disease phenotypes have all been described in patients with *CNTNAP2* LoF mutations[10,35] while corpus callosum volumetric

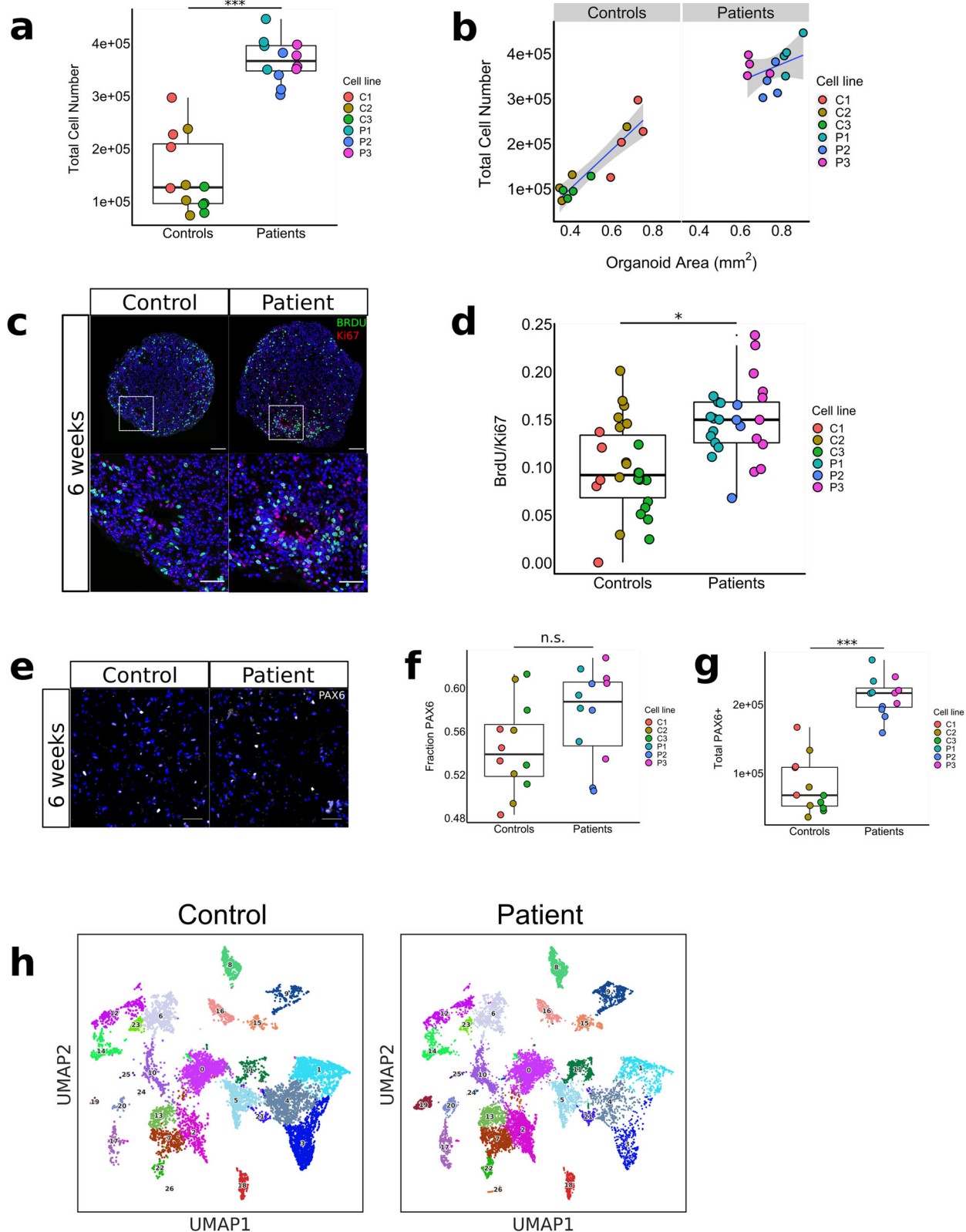

changes had not been investigated until now. In order to investigate the latter, we carried out a qualitative comparison of corpus callosum volumetric differences on MRI between the *CNTNAP2* mutation carriers and matched healthy controls without the mutation and found a decrease in corpus callosum volume in patients utilizing a one-sample *t*-test was performed between the size deviation for the templates (which are 0), and

found that mutation carriers indeed demonstrate a decrease in corpus callosum volume (*t*-test; $p = 0.0127$) (Supplementary Data 12).

**Site-specific repair of c.3709DelG mutation using CRISPR-Cas9 rescues cortical overgrowth phenotypes and transcriptional changes.** To confirm the causal relationship between the

**Fig. 3 Volumetric increase is driven by increased proliferation, which leads to expansion of progenitor pool and increased total cell number. a, b** Total cell number quantification of patient- and control-derived organoids after 6 weeks in culture, showing a 2.5-fold increase in total cell number for patient-derived organoids (LRT, $p = 0.00043$, $\chi2(1) = 13.25$, $n = 3$ cell lines/genotype, 4 orgs/cell line) (**a**), and the linear relationship between 2D organoid surface area and total cell number for patient- and control-derived organoids after 6 weeks in culture. (Controls: $F(1,10) = 50.22$, $R^2 = 0.83$, $p = 3.348e$-05; patients: $F(1,10) = 1.82$, $R^2 = 0.15$, $p = 0.21$, $n = 3$ cell-lines/genotype, 4 orgs/cell line). Boxplots in panel **a** display median, first and third quartiles, and whiskers showing the largest and smallest values no further than 1.5 times the inter quartile range from first and third quartile, respectively. *$p < 0.05$; **$p < 0.01$; ***$p < 0.001$. **c, d** Representative image of organoid sections stained for BrdU and Ki67 after 6 weeks in culture (**c**) and quantification (**d**) of ratio BrdU/Ki67 between patient- and control-derived organoids, showing a 1.5-fold increase in BrdU/Ki67 ratio for patient-derived organoids (LRT, $p = 0.02$, $\chi2(1) = 5.55$, $n = 3$ cell-lines/genotype, 3 orgs/cell line, 3 sections/org). Using the formula $Tc = Ts/(BrdU + /Ki67 + )$, where $Tc$ = cell cycle length, and $Ts$ = S-phase length, estimated mean cell cycle lengths for patient- and control-derived organoids are ~54 and ~83 h, respectively. Scale bars in **c** represent 100 μm in top panels, and 50 μm in lower panels. Boxplots in panel **d** display median, first and third quartiles, and whiskers showing the largest and smallest values no further than 1.5 times the inter quartile range from first and third quartile, respectively. *$p < 0.05$; **$p < 0.01$; ***$p < 0.001$. **e–g** Representative image of confocal microscopy images of nuclei from homogenized control and patient-derived organoids cultured for 6 weeks and stained for NPC-marker PAX6 (**e**), quantification of fractions of PAX6 + cells at 6 weeks in vitro (**f**) and total number of PAX6 + cells at 6 weeks in vitro (**g**), showing a 2.6-fold increase in total PAX6 + cells at 6 weeks (LRT, $p = 0.0003$, $\chi2(1) = 13.35$, $n = 3$ cell-lines/genotype, 4 orgs/cell line), with an equal fraction of PAX6 + cells at 6 weeks. Boxplots in panel **f** and **g** display median, first and third quartiles, and whiskers showing the largest and smallest values no further than 1.5 times the inter quartile range from first and third quartile, respectively. *$p < 0.05$; **$p < 0.01$; ***$p < 0.001$. **h** UMAP plots displaying the 26 clusters of cell population determined by the unsupervised Leiden clustering method for all control-derived organoids ($n = 2$ samples; 11,328 cells) (left panel) and all patient-derived organoids ($n = 3$ samples; 16,780 cells) (right panel). There is a lower number of cells in clusters 1, 3, and 4 in patient-derived samples, corresponding to cell types EN-V1, EN-PFC, and nEN-early annotated in Fig. 1D.

homozygous *CNTNAP2* c.3709DelG mutation and the cortical overgrowth disease phenotypes, we used clustered regularly interspaced short palindromic repeats (CRISPR)-Cas9[36] to generate an isogenic 'rescue' line by reintroducing the guanine base at position 3709 in one of the patient-derived hiPSC lines (Fig. S4A). Sanger sequencing of a CRISPR edited single-cell derived clone confirmed the homozygous wildtype DNA sequence (Fig. 5a). Western blotting confirmed the presence of the CNTNAP2 protein in the organoids derived from the CRISPR line, at levels similar to those of control-derived organoids (Fig. 5b). We subsequently repeated the experiments by following the same organoid differentiation protocol described above. Organoids generated from the CRISPR-rescue hiPSC line had a 1.5-fold reduction in 2D projected surface area compared to the organoids derived from the unedited patient line (Fig. 5c, d) (*t*-test, $n = 9$/line $p = 0.0014$) – which corresponds with the observed baseline difference in 2D surface area between patient- and control-derived organoids. In addition, the other overgrowth phenotypes, including increased total cell number (*t*-test, $n = 4$/line $p < 0.001$) (Fig. 5e) and increased PAX6 + progenitor pool at 6 weeks (*t*-test, $n = 4$/line, $p = 0.002$) (Fig. 5f), were rescued to comparable numbers observed in the organoids generated from healthy controls. At the bulk-transcriptional level, 576 genes were differentially expressed between organoids derived from the CRISPR-rescue line when compared to its parental patient-derived line. Of the 339 genes that were differentially expressed in patient-organoids, 67 of these were expressed in the opposite direction in the CRISPR-rescue line, indicating a partial rescue (Fig. S4B, C) (permutation analysis, $n = 10000$, $z = 18.6$, $p = 2.3e$-77). These 67 genes are enriched for biological processes related to neurodevelopment including cellular proliferation (Fig. S4D). Regardless of whether reaching the threshold for statistical significance, the gene expression directionality of the 339 DEgenes was generally reversed between patient- versus control lines and CRISPR-rescue versus patients line (Fig. 5g). PCA revealed a large batch effect in sample clustering represented by PC1 and a genotype-rescue effect represented by PC2 (Patient vs. CRISPR-rescue) (Fig. S4E, F). Of the 100 genes with the highest contribution to PC1 (86% of the variance) only three genes are present among the 339 DEgenes from the initial patient-control batch (permutation analysis; $n = 10,000$; $z = 0.99$; $p = 0.32$), whereas the top-100 contributing genes for PC2 (6% of the variance) include 41 of these 339 DEgenes (permutation analysis; $n = 10,000$; $z = 29.7$; $p = 8.86e$-

195), indicating the batch effect manifests through genes different from the 339 DEgenes from the initial patient-control batch (Fig. S4G).

## Discussion

In this study, we utilized forebrain organoids generated from hiPSCs derived from patients carrying the homozygous c.3709DelG mutation in *CNTNAP2* and healthy controls to investigate the effects of this mutation on cortical embryonic development. We show that *CNTNAP2* is most highly expressed in several types of PFC excitatory neurons. We discovered increased proliferation in PAX6-positive NPCs and other dividing cells, leading to an increase in the generation of total cell number. This increase in number of cortical cells is responsible for a corresponding increase in overall organoid volume in patient-derived organoids. GO analysis from both bulk RNAseq- and scRNAseq analysis corroborates abnormal cellular proliferation and differentiation. We further show that there is an enrichment of these differentially expressed genes for ASD-associated genes and WGCNAs associated with ASD and other neurodevelopmental disorders. The cell type-specific signature genes expressed in the affected neuronal types are associated with clinical disease phenotypes that have been described in patients with *CNTNAP2* LoF mutations. Finally, by repairing the pathogenic mutation using CRISPR-Cas9, we were able to rescue these cortical overgrowth phenotypes, thereby confirming a causative effect of the homozygous LoF mutation in the context of an identical genetic background.

The main findings from this study confirm that the homozygous c.3709DelG mutation in *CNTNAP2* leads to abnormal brain development in processes analogous to those occurring during the first trimester of embryonic development. Thus, the findings support the prevailing hypothesis of an early embryonic origin of the pathophysiology of ASD[37]. This increase in total cell number could be related to the increased cortical thickening seen in ASD, as was also been observed on MRI of patients with *CNTNAP2*-associated ASD. More recently, other studies deploying hiPSC-derived neuronal culture techniques have also reported increased NPC proliferation and aberrant neurogenesis in patients with ASD. This includes both monogenic forms of ASD[38] and idiopathic ASD[8,39] – in both 2D and 3D neuronal cultures. These findings could be causally related to two clinical phenomena observed in patients with ASD that may have at least

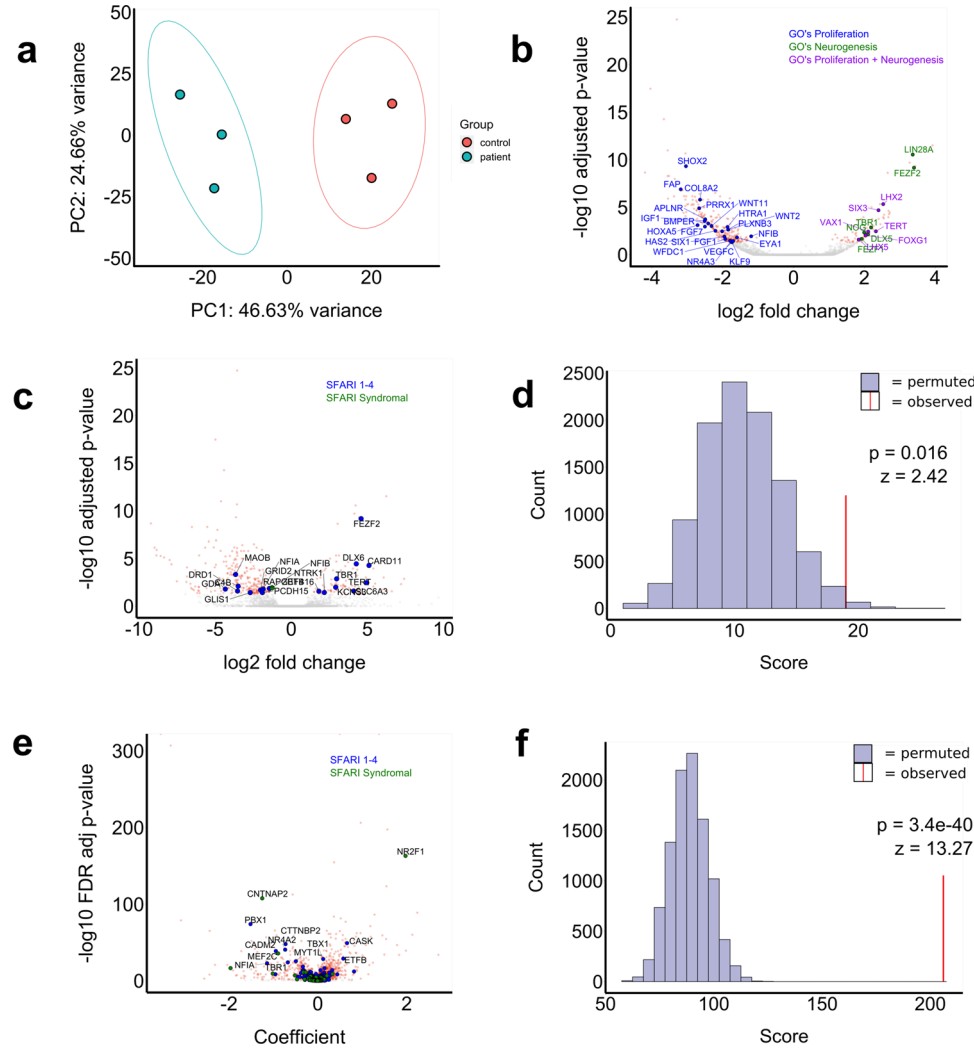

**Fig. 4 Differentially expressed genes are enriched for GOs related to cell proliferation and neurogenesis, as well as ASD-associated genes. a** Principal component analysis shows a clear distinction between the expression profiles of patient-compared to control-derived organoids at 8 weeks in vitro. **b** Volcano plot showing differentially expressed genes between patient- and control-derived organoids at 8 weeks in vitro. Genes reaching statistical significance are colored in red, and genes below statistical significance threshold are colored in gray. A total of 250 genes are downregulated while 89 are upregulated. Genes are highlighted according to their Gene Ontology (GO). GOs related to cell proliferation (blue) (GO:0050673, GO:0050678, GO:0010463, GO:0010464, GO:0050679), neurogenesis (green) and both proliferation and neurogenesis (purple) (GO:2000177, GO:0072091, GO:0061351, GO:1902692, GO:0072089, GO:2000179, GO:0007405, GO:0050768, GO:005076). **c, d** The same volcano plot shown in panel **b**, but now with ASD-associated genes curated by SFARI enriched among the DE genes highlighted. Blue dots are genes in SFARI categories 1–4, green dots are in category 'syndromic' (**c**). Statistical significance was tested using a permutation test ($n = 10,000$) showing an enrichment higher than enrichment expected from random events ($z = 3.28$, $p = 0.001$). **e, f** Volcano plot showing genes that are differentially expressed in three *CNTNAP2*-expressing cell types (EN-V1, EN-PFC, and nEN). Genes reaching statistical significance are colored in red, and genes below statistical significance threshold are colored in gray. Blue dots are genes in SFARI categories 1–4, green dots are in category 'syndromic. Of 2800 differentially expressed genes in these cell types, 215 are present in the SFARI database (**e**). Enrichment for SFARI genes at the single-cell level was more pronounced than in the bulk RNA sequencing analysis as indicated by the bar plot displaying the permutation analysis ($n = 100,000$, $z = 13.93$, $p = 4.1e-44$).

a partial overlapping etiology: early brain overgrowth and an increase in head circumference or macrocephaly (defined as head circumference >97th percentile). A recent systematic review and meta-analysis that analyzed 27 studies reported macrocephaly in 15.7% of ASD cases 3% in controls[40]. The same study also reported an increase in total brain volume (defined as brain volume 2 standard deviations above the mean) in 9% of patients with ASD.

Of note, the overgrowth phenotype described in this study has not been recapitulated in *Cntnap2* null mice[16,41]. One study quantified the volume of both medial prefrontal and somatosensory cortices, and found no difference between null- and wildtype mice[41]. Thus, although mouse models can capture some

of the clinical features of a human condition such as ASD, important aspects of such a disease may be missed since these may exclusively be present in humans. Given the fact that one of the defining features of the human brain compared to other mammals is its expanded gyrencephalic neocortex[42] that is thought to give rise to our abilities of higher order brain processes, such as cognition and language, it is not entirely surprising that a truncation of a protein with a critical role in cortical development could have different phenotypic outcomes in different mammalian species. Species-specific differences in the role that *CNTNAP2* plays in brain development is reflected in its differential expression pattern as well. This was shown using in situ *hybridization*, revealing that gene expression is more

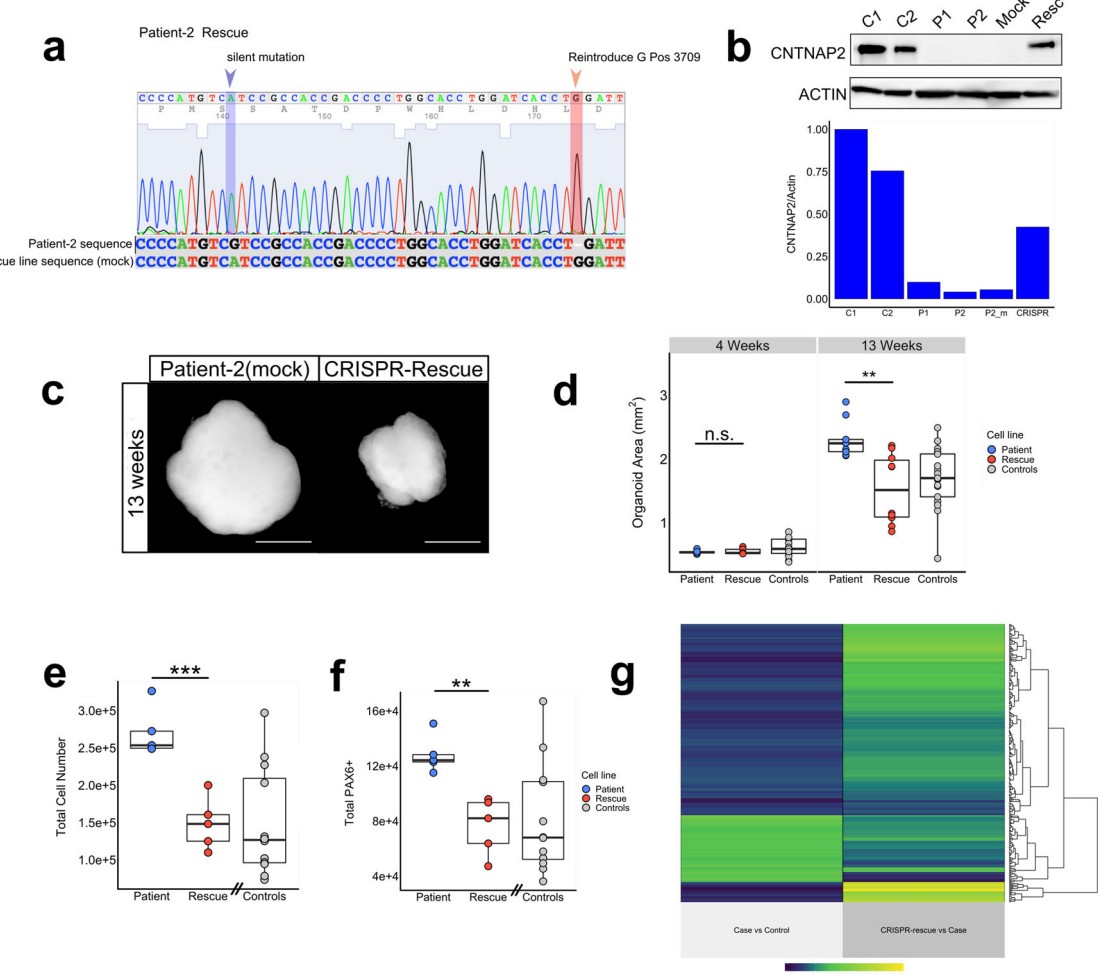

**Fig. 5 CRISPR-mediated repair of pathogenic c.3709DelG mutation rescues overgrowth phenotypes. a, b** Sanger sequencing trace of hiPSC-line derived from patient-2 after CRISPR-Cas9 genome editing. The rescued line shows a guanine base introduced at cDNA position 3709 and a silent blocking mutation at position 3675. Sequence is aligned to original patient-2 DNA sequence (**a**). Western blotting confirms the presence of CNTNAP2 protein in organoids generated from the CRISPR-rescued line at a similar level as control lines, and absence of the protein in the parental patient-2 cell line-derived organoids (**b**). Source data are provided as a Source Data file. **c, d** Representative image of bright field microscopy images of organoids derived from patient-2 and CRISPR-rescued line at 13 weeks in vitro (**c**), and corresponding quantification showing a 1.5-fold decrease in area for the CRISPR rescue line compared to the unedited patient line (**d**) (two-sided $t$-test, $t(14.1) = 3.93$, $p = 0.0014$, $n = 9$ orgs/cell line). Gray dots represent pooled control data points also displayed in Fig. 2a. Scale bar represents 500 μm. Boxplots in panel **d** display median, first, and third quartiles, and whiskers showing the largest and smallest values no further than 1.5 times the inter quartile range from first and third quartile, respectively. *$p < 0.05$; **$p < 0.01$; ***$p < 0.001$.
**e, f** Quantifications of total cell number at 6 weeks ($t$-test, $t(7,98) = 5.68$, $p < 0.001$, $n = 5$ orgs/cell line) (**e**), total number of PAX6 + cells at 6 weeks (two-sided $t$-test, $t(6,87) = 4.68$, $p = 0.002$, $n = 5$ orgs/cell line) (**f**). Total cell number, and total number of PAX6 + cells are decreased in the CRISPR rescue-line compared to the parental patient line, confirming the causal effect of the single mutation on the observed phenotypes. Gray dots represent pooled control data points also displayed in Fig. 3a. Boxplots display median, first, and third quartiles, and whiskers showing the largest and smallest values no further than 1.5 times the inter quartile range from first and third quartile, respectively. *$p < 0.05$; **$p < 0.01$; ***$p < 0.001$. **g** Heatmap displaying the log2 fold-change of the 339 genes differentially expressed between control- and patient-organoids (left column) and between the CRISPR-rescue line versus its parental patient-derived control (right column). The inversion of the expression direction for the majority of the 339 genes highlights that the DE gene expression-level is largely rescued in the CRISPR line derived organoids.

diffusely distributed throughout the entire brain in rodents while there is a more focal expression in humans with a preferential localization in prefrontal cortex[23].

The findings from this study lead to interesting questions regarding the development of novel treatments for neurodevelopmental disorders, such as *CNTNAP2*-associated ASD. It is critical to determine the extent to which pathophysiological processes that occur as early as in the first trimester of pregnancy remain reversible after birth. Are the cortical overgrowth phenotypes critical to cognitive- and behavioral deficits seen in these patients, and if so, during which time window should treatments

aim to target these early developmental processes? In this study, we show that the field of brain organoid modeling holds great promise for working towards a preclinical modeling system with face validity, with the ultimate goal of creating targeted treatments that may eventually improve the lives of patients with severe neurodevelopmental disorders such as *CNTNAP2*-associated ASD.

## Methods

**Ascertainment of subjects and clinical phenotypes**. The study was approved by the Lancaster General Hospital (LGH) Institutional Review Board (IRB) (LGH IRB

protocol number 2008-095). All investigators complied with all relevant ethical regulations for work with human participants. Parents of all subjects provided written informed consent to participate on behalf of their children. Neuropsychiatric diagnoses and head circumferences measurements were done during routine clinical visits for pediatric care at the Clinic for Special Children in Strasburg, Pennsylvania. The generation of hiPS cell lines and derived forebrain organoids was approved by New York State Psychiatric Institute (NYSPI) IRB (NYSPI IRB protocol number 7500) and Columbia University's Human Embryo and Embryonic Stem Cell Research Committee.

### MRI analysis

*Gray matter volume.* We quantified total gray matter volume relative to the total brain volume of 6 whole-brain MRIs from patients carrying the c.3709DelG mutation in *CNTNAP2*, and compared these to 4 template MRIs that were generated from whole-brain MRIs of age-matched healthy children[43,44]. We used Statistical Parametric Mapping for image segmentation and summed gray matter and white matter masks to calculate total brain volume. For each image, the sum value of voxels for each tissue class was used to generate a ratio between gray matter and white matter.

*Corpus callosum volume.* Clinical MRI scans were spatially normalized into the corresponding age-matched template space T1-weighted MRI scan. Since all scans were of varying thicknesses, with axial acquisitions preventing accurate fine quantification of the corpus callosum, a side-by-side qualitative evaluation was made for each subject and template scan. For each participant MRI scan in corresponding template space, a number was assigned to indicate "corpus callosum size deviation from normal" with scores of 0, 1, and 2 (no change, subtle change, marked change) and the sign being direction (+ was template larger than control). This was done three times for each participant blindly and the median score recorded. A one-sample *t*-test of the values and the template values (which are 0) was performed.

### hiPSC generation and characterization

hiPSC lines were generated from three patients with *CNTNAP2*-associated ASD carrying the homozygous c.3709DelG mutation and three healthy controls from the same Amish population who did not carry mutation, by a non-integrating Sendai virus-based reprogramming method as previously described[45]. Karyotyping was performed on twenty G-banded metaphase cells at 450–500 band resolution as previously described[46]. All hiPSC lines used in this study were between passage 10 and 25. The difference in passage number for each pair of patient- and control-derived line never exceeded three passages.

### Forebrain organoid culture

Forebrain organoids were generated from patient- and control hiPSC lines using a previously published organoid differentiation protocol[47], with minor modifications. Briefly, hiPSCs were dissociated with 0.5 mM EDTA in PBS and triturated to generate a single-cell suspension. A total of 10,000 cells were then plated into each well of an ultra-low-attachment 96-well plate (Nunc) to form embryoid bodies (EBs) in medium containing mTesR1, 1 μg/ml heparin and Penicillin-Streptomycin antibiotics. EBs were exposed to medium containing the ROCK inhibitor Y27632 (50 μM) for the first 24 h, followed by 5 days without interference in 96-well plates. On day 5, the medium was switched to medium containing DMEM/F12 (1:1) (Gibco; 11330), N-2 supplement (Gibco; 17502-048), MEM-NEAA (Invitrogen), Glutamax (Invitrogen 35050-061), Penicillin-Streptomycin and 0.1 mM ®-Mercaptoethanol, Dorsomorphin (2 μM), SB431542 (10 μM) and IWR1e (3 μM). On day 25, 0.2% Chemically Defined Lipid Concentrate was added to this medium until day 39. On day 40, EBs were transferred to ultra-low attachment 24 well plates with 1 ml of medium per organoid/well. Between day 40 and 80, 1 ml of medium containing DMEM/F12 (1:1) (N-2 supplement MEM-NEAA, Glutamax, Penicillin–Streptomycin and 0.1 mM ®-Mercaptoethanol, 10% fetal bovine serum (FBS), and 1% Matrigel (Corning) was replaced weekly. After day 80 medium was replaced once per week. Matrigel was added bi-weekly to prevent excess Matrigel deposition on the outer shell of the organoid.

### Immunohistochemistry

See Supplementary Data 1 for all antibodies and respective concentrations used. Organoids were fixed for 4 h in 10% formalin at room temperature, cryo-protected in 15% sucrose, dissolved in PBS for 4 h, and then placed in 30% sucrose in PBS overnight. The next day organoids were embedded in optimal cutting temperature (O.C.T.) solution and cut into 14 μm sections on a Leica Cryostat. After heat-induced epitope retrieval in 40 mM Sodium Citrate/ 0.05% Tween-20 (pH6), tissue sections were blocked for 1 h in 10% horse serum in PBS containing 0.1% TritionX-100 (PBSTx0.1). Primary antibodies were incubated overnight at 4 °C in PBSTx0.1. Secondary antibodies were incubated for 1 h at room temperature and nuclei were counterstained with 300 nM DAPI solution. Sections were mounted on glass slides using Prolong antifade solution and imaged on a LEICA SP8 confocal microscope.

### Western blotting

Proteins were extracted in lysis buffer containing protein inhibitor (cOmplet Mini, Roche, Ref: 11836170001) in tissue protein extraction reagent (Thermo Scientific, Ref: 78510) according to manufacturer's protocol. Briefly, organoids were washed twice with DPBS, then incubated in lysis buffer for 10 min on ice and vortexing vigorously. Samples were spun down at 10,000×*g* and stored at −80 °C for downstream analysis. Extracted protein was separated on a NuPAGE 4–12% gradient gel (Invitrogen, Ref: NP0322). Primary antibodies (see Supplementary Data 1) were bound by HRP-conjugated secondary antibodies and visualized using the Azure c600 imaging system.

*Light-sheet microscopy.* Whole organoids were fixed and stained in 300 nM DAPI in PBS for 30 min to ensure staining of the nuclei of the outer shell of the organoid. Light-sheet microscopy images were acquired on a Zeiss Z-1 Light-sheet microscope from 6 angles in 45° degree increments starting at 0°.

*gRNA and donor template design.* Guide-RNAs were designed using *CT-finder* software[48]. gRNAs were designed around the c.3709DelG mutation site in the *CNTNAP2* locus on chromosome 7q35. gRNA target A on the minus strand is 5′-GGGTCGGTGGCGGACGACATGGG-3′ and gRNA target B on the plus strand is 5′-GCACCTGGATCACCTGATTCAGG-3′. The donor template was a single stranded 181-bp oligonucleotide (see Supplementary Data 2). This sequence is complementary to the antisense DNA strand and contains the wildtype DNA sequence and a silent blocking mutation near the 5′-end of gRNA target A. The donor template flanks the mutation site with 90 bp on each side (Fig. S2A).

### hiPSC gene-editing using CRISPR Cas9

We randomly selected one of the three patient hiPSC lines for CRISPR genome editing. We followed a CRISPR Cas9 gene editing protocol previously described[49] with minor modifications. Briefly, gRNA oligonucleotides were cloned into the plasmid pSpCas9n(BB)−2A-GFP encoding Cas9-nickase. pSpCas9n(BB)−2A-GFP (PX461) was a gift from Feng Zhang (Addgene plasmid # 48140; http://n2t.net/addgene:48140; RRID:Addgene_48140). hiPSCs were maintained in MTesR1 medium. Cells were passaged using 0.05 mM EDTA in PBS or Accutase. Prior to nucleofection, cells were treated with ROCK-inhibitor for 24 h. Plasmids and donor template were introduced into the hiPSCs through nucleofection using Amaxa nucleofector II. In all, 72 h post nucleofection, GFP + cells were selected using FACS and grown as single-cell-derived colonies onto irradiated MEF coated 35 mm dishes. Colonies with undifferentiated hiPSC morphology were manually picked and further grown on 96-well plates. Clones derived from single cells were genotyped using Sanger sequencing on PCR amplicons (see Supplementary Data 2 for primers).

### Viral transduction

pAAV.hSyn.eGFP.WPRE.bGH was a gift from James M. Wilson (Addgene plasmid # 105539; http://n2t.net/addgene:105539; RRID: Addgene_105539). The virus was added to the organoid growth media and incubated for 1 week at a concentration of $10^7$ virus particles per ml. Transduction efficiency was evaluated 10 days post-transduction using an epifluorescence microscope.

### Isotropic fractionation

We adapted a previously developed protocol for to accurately calculate total cell number and relative cell type fractions based on nuclear immunostaining[24]. Briefly, organoids were fixed for 4 h in 10% formalin at room temperature. After blocking for 4 h in 10% horse serum in PBS containing 0.2% Triton X-100 (PBSTx0.2), primary antibodies were incubated PBSTx0.2 on whole organoids for 20 h on a shaker at room temperature to ensure complete tissue penetration. Organoids were washed for 4 h in PBSTx0.2. Secondary antibodies were incubated for 20 h in in PBS PBSTx0.2, followed by a 4-h wash in PBSTx0.2. Organoids were fixed for a second time for 20 h, and then homogenized using a glass tissue homogenizer in 10 mM sodium citrate buffer containing 1% Triton X-100. Total cell counts were performed by counting DAPI-stained nuclei in a hemocytometer on an upright Zeiss epifluorescence microscope. Relative cell type fractions were calculated from images acquired on a Leica SP8 confocal microscope.

### BrdU labeling

Organoids were incubated using 10 μM BrdU dissolved in growth medium for 2 h at 37 °C, then washed twice with PBS and subsequently fixed for 4 h in 10% formalin at room temperature. We subsequently followed our immunostaining protocol to stain for BrdU after heat denaturing the DNA in a 40-mM sodium citrate buffer containing 0.05% Tween-20. For the BrdU/Ki67 quantification analysis, serial sections of 14 μm were made every fifth section, with 3–4 sections for each organoid.

### Total RNA isolation and bulk RNA sequencing

Ten forebrain organoids per line (i.e., 3 control-derived lines, 3 patient-derived lines) were collected at 8 weeks in vitro. Total RNA was isolated from hiPSC-derived cortical neurons using miRNeasy kit (Qiagen, USA) according to instructions of the manufacturer. RNA was suspended in RNase-free water. The concentration and purity of each sample was determined by spectrophotometer (ND1000; Nanodrop) and confirmed by Microchip Gel Electrophoresis (Agilent) using Agilent 2100 Bioanalyzer Chip

according to the instructions of the manufactures. A poly-A pull-down step was performed to enrich mRNAs from total RNA samples (200 ng to 1 μg per sample, RIN > 8 required) and proceeded on library preparation by using Illumina TruSeq RNA prep kit. Libraries were then sequenced using Illumina HiSeq2000 at the Columbia Genome Center. Multiplex samples with unique barcodes were mixed in each lane, which yields targeted number of single-end 100 bp reads for each sample, as a fraction of 180 million reads for the whole lane. RTA software (Illumina) was used for base calling, and bcl2fastq (version 1.8.4) for converting BCL to fastq format, coupled with adapter trimming. The reads were mapped to a reference human genome (hg19) using Tophat2.0 (version 2.0.4) with four mismatches (--read-mismatches = 4) and 10 maximum multiple hits (--max-multihits = 10). We have uploaded the bulk and single-cell RNAseq data used in this paper in NCBI's Gene Expression Omnibus under accession code GSE174569.

**Transcriptome correlation analysis**. Pearson's $r$ correlations were calculated between the top 1000 most highly expressed genes from the bulk RNA sequencing data from the three control-derived organoid lines and the same genes from all available samples from the Brainspan transcriptome atlas[21].

**Differential expression analysis**. DESeq2 (Version 1.28.1), an R package based on a negative binomial distribution that models the number of reads from RNA-seq experiments and tests for differential expression[50], was used to determine differentially expressed genes (DEGs) between mutants and control samples. The list of significantly DEGs was defined at false discovery rate (FDR) adjusted $p$-value (padj) < 0.05.

**Gene ontology analysis**. To determine a common functional relationship among the top DEGs, the enrichment of biological processes was tested using Metascape[26] with default settings. Gene lists of upregulated and downregulated DEGs were analyzed separately.

**Permutation tests**. Permutation tests were performed by generating 10,000 lists of gene-sets with equal size to the differentially expressed genes. We then compared these randomly generated genesets to the gene lists of interest (WGCNA networks, ASD-SFARI genes (SFARI-Gene_genes_10-31-2019release_10-05-2020export) as well as control gene lists derived from other common diseases, and tested whether the overlap between DEGs and permuted gene lists differed in a statistically significant manner from the randomly generated gene lists. Two-sided $p$-values were derived from z-scores using formula $z = (x - \mu)/\sigma$.

**Image analyses**. Confocal- and light-sheet microscopy images were processed using ImageJ software (Version 2.0.0-rc-69/1.52p) (https://imagej.nih.gov/ij/). BrdU/Ki67 quantifications and nuclear fraction quantifications were carried out using Cell Profiler software (Version 4.1.3)[51].

**Statistical analyses**. All statistical and bioinformatic analyses were performed using $R$ programming language (version 4.0.0)[52]. The graphs in the manuscript display boxplots showing median, first-, and third quartiles. Whiskers show the largest and smallest values no further than 1.5 times the interquartile range from the first and third quartile, respectively, with all data points plotted individually. Comparisons between patient- and control-derived organoids were done using a linear mixed model from software package $Lme4$[53] with genotype status designated as a fixed effect and cell line as a random effect. Relationships between two continuous variables were tested using linear regression. Statistical significance from the mixed model was calculated using Likelihood Ratio Tests (LRT). For comparisons of the CRISPR rescue-line to its original patient line, and the gray matter volume analysis, we used two-sided Student's $t$-tests. $P$-values < 0.05 were considered statistically significant. Outliers were never removed from any of the analyses.

**Single-cell RNAseq dissociation and sequencing**. We collected three organoids per samples and dissociated using the papain dissociation protocol (Worthington Biochemicals) according to the manufacturer's protocol. Briefly, organoids were incubated for 30 min at 37 °C in Earle's Balanced Salt Solution (EBSS)/Albumin-ovomucoid inhibitor/papain with gentle shaking and mechanical dissociation. The pellet was resuspended and incubated with (EBSS)/Albumin-ovomucoid inhibitor/DNAse for 2 min followed by incubation with ovomucoid for 2 min. The cells were washed and resuspended with DMEM/F12. After determination of the cell density and viability, cells were submitted to single-cell RNA sequencing (10X Genomics, Chromium Single cell 3' v3) to recover ~5000 sequenced cells per sample with more than 50,000 reads per cell. We have uploaded the RNAseq data used in this paper in NCBI's Gene Expression Omnibus under accession code GSE174569.

**Single-cell RNAseq data analysis**. The Cell Ranger 4.0.0 pipeline (10x Genomics) was used to align and aggregate all reads from scRNA-seq to the GRCh38 human reference genome with default settings and produce a balanced "filtered feature bc

matrix". The data was imported into the SCANPY software (Version 1.7.1)[54] where genotype information was added. Quality control was conducted to ensure all samples analyzed contained highly consistent quality. Cells expressing a minimum of 200 genes were kept, and counts were normalized for each cell by the total expression, multiplied by $10^6$ and log-transformed. LIGER (Version 2.0.1) software[55] was used to integrate the different samples and variation in the cells' transcriptional profile was visualized by uniform manifold approximation and projection for dimension reduction (UMAP) function[56] in SCANPY. To annotate the cell types in brain organoids, we used two independent approaches. First, we downloaded the gene-cell count matrix and cell type annotation files of the single-cell RNAseq analysis of human embryonic prefrontal cortex tissue from the cell browser[22]. We then utilized this dataset to train the SingleR software(Version 1.0.6)[57], and to infer the cell types in our dataset. In addition, we used the "Leiden clustering method" implemented in the SCANPY to conduct unsupervised cell population clustering[58]. We used the Rank "rank genes groups analysis" of SCANPY to determine the top 100 cell type differentiating genes of each cell type or each cell cluster annotated by singleR and Leiden clusters. For each cell type or cluster, differentially expressed (DE) genes were determined using model-based analysis of single-cell transcriptomics (MAST, Version 1.12.0) algorithm[59]. Although our correlation annotation approach indicated the presence of a small set of endothelial and microglia cells, our further analysis using predefined biomarkers (adapted from Suppl Table 2 in ref. [25]) indicated that the expression of key biomarkers of these cell types are either absent or weak as compared to other neural types (Suppl Fig. 1J). Therefore, we omitted these cell types for further downstream analysis.

**Reporting summary**. Further information on research design is available in the Nature Research Reporting Summary linked to this article.

## Data availability
All data are available from the authors upon request. Bulk RNA sequencing transcriptome data are available under accession code: GSE174569. Publicly available data used in this paper includes: - Brainspan transcriptome database (Miller et al.[21]) www.brainspan.org - human embryonic PFC single-cell transcriptome (Nowakowski et al.[22]) www.sciencemag.org/content/358/6368/1318/suppl/DC1 - human embryonic PFC cell type markers (Bhaduri et al.[25]) www.nature.com/articles/s41586-020-1962-0#Sec38 Figures associated with available raw data in the source data file: Figs. 1E, F, 4A–F, 5B, G, S1J, S3A–P, and S4B–G In the other figures, all datapoints have been plotted individually. Source data are provided with this paper.

## Code availability
All code used is available from the authors upon request.

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

## Acknowledgements

The authors would like to thank Dr. Yuanjia Wang and Chen Chen for their advice on the statistical analyses using the linear mixed model. We thank Dr. Maura Boldrini for her advice on the BrdU labeling experiments. We thank Drs. Alexander Sosunov, Andrew Dwork and Gorazd Rosoklija for their advice on histology experiments. We thank Joseph Sall from NYU Langone's Microscopy Laboratory (supported by the Cancer Center Support Grant P30CA016087 at the Laura and Isaac Perlmutter Cancer Center) for technical assistance with the light-sheet microscopy image acquisition. We thank Luis Aparicio de Santiago for his contribution to the scRNAseq analysis. We thank Julie Shabto, Qanetha Ahmed, Yelizaveta Gribkova, and Huber Rodriguez-Tejada for their contribution to western Blotting and imaging experiments. We thank Dr. Theresa Swayne and Dr. Laura Munteanu for their advice on image processing and analysis at the Confocal and Specialized Microscopy Shared Resource of the Herbert Irving Comprehensive Cancer Center at Columbia University, supported by NIH grant #P30 CA013696 (National Cancer Institute). This research was partially supported by Irving Institute/CTO Pilot Award (UR007953) to B.X.; B.X. and S.M. are supported by NCAT UG3 NS115598.

## Author contributions

S.M. and B.X. conceived the research project, J.O.J., M.G., H.Z., C.L., Y.S., K.S., and G.C. performed experiments and collected data, J.O.J., B.X., B.M., M.G., F.P., and G.C. analyzed data, B.C. reprogrammed patients' samples, S.M., K.S., and K.B. were involved in the recruitment of the patients. S.M., B.X., C.K., J.A.J., S.K., and J.O.J. contributed to design of the research project and discussion of the data. J.O.J., S.M., B.X., J.A.J., and C.K. wrote the manuscript with input from the other authors.

## Competing interests

The authors declare no competing interests.
