## [Peer Review File · Nature Communications]

Reviewers' comments:

Reviewer #1 (Remarks to the Author):

In the manuscript entitled "Cortical Overgrowth in a Preclinical Forebrain Organoid Model of CNTNAP2-Associated Autism Spectrum Disorder", Jong et al. report a patient-derived organoid model of CNTNAP2 mutation. The authors generate forebrain organoids from patients carrying a homozygous c.3709delG mutation in CNTNAP2, and show that the organoids exhibit an increased volume that is consistent with the increase in gray matter volume observed in patients. They determine that the increased volume stems from the increased proliferation of neural progenitor cells, and show that this phenotype is rescued upon correction of the mutation by CRISPR-mediated gene editing. In addition, the authors show that gene expression is dysregulated in the patient-derived organoids, and determine that the differentially expressed genes include known ASD-associated genes and are enriched for genes in ASD-associated weighted gene co-expression networks.

The data is interesting, and has clinical relevance. However, the authors should consider characterizing their organoid model in more detail. There are a few points that should be addressed to strengthen the study:

1. The authors immunoblot for CNTNAP2 in 0-10 week-old organoids (Fig. 1C), and suggest that CNTNAP2 expression in their organoids is in concordance with in vivo expression patterns, making their organoids a good preclinical neuronal model system (Lines 398-400). To make a stronger argument for this, the authors should directly compare their organoids to the Brainspan data to establish: (1) What brain region(s); (2) What fetal age their organoids most closely correspond to. Given that CNTNAP2 expression varies across brain regions and developmental timepoints (Fig. S1J), it is important to confirm that the 6-13 week organoids match the brain regions and developmental timepoints of interest, making them an appropriate disease model. This comparison could be done by performing correlation analysis on the transcriptomes of the organoids and the Brainspan samples used to generate Fig. S1J. Please also include a scale for the heatmap colors in Fig. S1J.
2. The authors should note if there were any electrophysiological differences when they performed whole-cell patch-clamp recordings on the 8 week patient- and control-derived organoids (Lines 370-373; Figs S1G-H), given that altered network activity has been reported in the brains of *Cntnap2* knockout mice (e.g. Lazaro et al., Cell 2019) as well as patient-derived CNTNAP2 mutant neurons. If differences are observed in patient- and control-derived organoids, the authors should then record from the CRISPR-rescued organoids to ensure that any phenotype is restored to the wildtype condition.
3. Along these lines, did the authors observe any additional phenotypes in their patient-derived organoids beyond the increase in total volume? CNTNAP2 mutations have been reported to be associated with a myriad of different phenotypes beyond macrocephaly (e.g., increased astrogliosis, delayed NPC migration), and it would be good for the authors to discuss if additional phenotypes are observed in their organoids beyond an increase in total volume. The authors do report observing altered gene expression in their patient organoids, but they should elaborate further on this (see point 5 below).
4. The authors should examine their RNA-seq data for the presence of CNTNAP2 mRNA transcripts. If no transcripts are detected in the patient organoids, this could suggest that the mutation destabilizes the mRNA by inducing nonsense mediated decay, thus accounting for

the lack of detectable protein (Line 388, Fig. 1D/S1I).

5. Along these lines, were there any differences observed in the expression of Kv1.1 and Nav1.2 channels in the RNA-seq data from the patient organoids? Altered expression of ion channels has been previously reported in the brain samples of human patients. It would be good to assess if this difference is also observed in the patient-derived organoids, and if it can be corrected in the CRISPR-rescued organoids.

6. In Fig. 3F, the 13-week patient organoids show a significantly lower fraction of PAX6+ cells as compared to control organoids ($p = 0.0008$). However, in Fig. 4H, the 13-week control and patient organoids do not appear to be significantly different in their respective fractions of PAX6+ cells (blue dots vs gray dots). Could the authors comment on this discrepancy?

7. The authors should specify if the CRISPR correction occurred on both alleles. If possible, the locus should be deep sequenced (e.g., amplicon sequencing) to ensure that no mutant allele is present. In addition, it would be good to assess what the level of rescue is relative to wildtype control. This could be done by updating the immunoblot in Fig. 4B to show the amount of CNTNAP2 protein in control, patient-2 and patient-2 rescue organoids.

8. To improve the flow of the manuscript, the authors should consider discussing the RNA-seq data before CRISPR rescue (i.e., switching Figs. 4 and 5 around). In this way, they could completely characterize their organoids before discussing rescue by gene editing. They should also consider performing RNA-seq in their rescued organoids to determine if the altered gene expression patterns are restored to wildtype levels.

9. Please review the numbering of all figures. There are a number of inconsistencies, for instance Figs. S1F-H are missing, Fig. S2 does not correspond to what is listed in the text (Line 418), and Fig. 4H is not listed in the legend.

Reviewer #2 (Remarks to the Author):

This paper describes the phenotype of syndromic CNTNAP2 mutations--which are associated with marked phenotypic abnormalities--on the growth of iPSC-derived organoids. Notably, these organoids display marked volume increases, initial progenitor hyperproliferation, and increased neurogenesis among other abnormalities. Importantly, knockout mice don't recapitulate these phenotypes nor do monolayer cultures.

The authors have written a very clear manuscript and the experiments are well designed. However, despite the remarkable phenotype, most of the experiments are "by the book" and little mechanistic investigation is done despite the nice CRISPR rescue. For example, some type of clonal labeling could clarify the natural history of cell expansion between controls and patients. Or the critical mediators downstream of the mutation could have been explored functionally. Also of note is the relative rarity of this particular gene mutation in ASD. The potential limitations of organoids should be discussed in terms of later synaptic events, compensatory events, and other things that can't be observed in such "young" organoids (compared with a neonatal or toddler brain).

Reviewer #3 (Remarks to the Author):

Manuscript # NCOMMS-19-26994-T

Cortical Overgrowth in a Preclinical Forebrain Organoid Model of CNTNAP2-Associated Autism Spectrum Disorder.

In this study, de Jong et al. investigate effects of homozygous c.3709DelG mutation of CNTNAP2 by using forebrain organoids generated from iPSC derived from patients from the Old Order Amish community carrying this mutation, that present with an increased head circumference and increased gray matter volume. The authors observed a similar volume increase in patient-derived organoids, that they linked to an enhanced proliferation of neuronal progenitors, that leads to an increase in the generation of cortical neurons and non-neuronal cell types. Remarkably, the observed phenotypes were rescued by correcting the CNTNAP2 gene mutation.

This work aims to demonstrate that organoids can be used to understand critical roles of genes associated with ASD. However, the overall conclusions are overstated and not supported by appropriate scientific evidence. Additional analysis needs to be performed to confirm these findings.

Major concerns.

1. The protocol for organoid generation is based on the method previously described by Kadoshima et al. 2013, however it introduces some modifications, e.g. embryoid bodies are cultured for 5 days in mTESR1 before neural differentiation is induced (through TGF β and WNT inhibition). The effect of such modifications on the cell types generated by using the originally described protocol are not known. A detailed analysis (e.g. by using immunohistochemistry) for different cortical markers across time points, needs to be included to help understand how the cellular composition changes over time and to support the choice of using organoids cultured for either 6 or 13 weeks, to underline cell-type phenotypes associated with disease.
2. One of the big findings of the manuscript is that patient- and control- derived forebrain organoids can recapitulate differences observed in patients, such as increased volumes. How the increased volume ratios change over time needs to be investigated further. At least, include analysis of volumes at week 6, that is when the highest increase in cell number is observed. IHC analysis at the same time points also have to be included (see point 1).
3. It has been previously shown that the organoid models are plagued by poor reproducibility (Quadrato et al. Nature 2017). The authors do not show or discuss the degree of reproducibility of cell type generation by using their organoid model. Unless data showing high levels of reproducibility of cell types generated by individual organoids are provided, a larger number of organoids need to be analyzed to support the findings.
4. One of the major findings of the manuscript is that patient-derived organoids show an increase in volume. However, the images in figure 3C show actually the opposite (mutant organoids are smaller than control). Are the images representatives of the sample?
5. Conclusions about progenitor proliferation and premature neurogenesis are exclusively

based on immunohistochemistry data obtained by using a limited number of markers (PAX6, Ki67 and NeuN). However, the assessment of single markers is not sufficient to define cell types. Single cell RNA-seq analysis would be the preferred method to define specific gene signatures that identify cell types and importantly, to claim that those are affected by the CNTNAP2 mutation. However, if this is not possible, at least analysis of a larger number of markers, to assess different cell types and biological processes, has to be included to support the conclusions the authors make.

6. The authors claim that forebrain organoids serve as preclinical neuronal model system to study the effects of homozygous LoF mutations on early human cortical development, based on the observation that the CNTNAP2 expression pattern in organoids is in concordance with in vivo gene expression. However, this concordance is based on comparison between protein levels assessed by immunoblotting (organoids) and gene expression assessed by bulk transcriptome analyses (human samples). If the authors want to make this claim, they need to investigate the similarity of cell types in their organoid system compared to that found in the developing human brain, at single cell level, as done in previous works (Camp et al. 2015). Otherwise, the authors need to resize this statement.

7. Previous work showed that Cortical Dysplasia Focal Epilepsy (CDFE) is more common in males (Ortiz-González et al. 2013). Instead, all the data presented in the current study are generated by using organoids derived from female patient iPSc. Is there any reason why the authors decided to focus on cell from female patients? Do they expect to find similar results in organoids derived from male patient iPSCs?

8. Fig. S1F, G and H are missing.

9. The authors claim that organoids from both patients and controls show features of mature neurons (data from Fig. S1F-H, that are missing in the manuscript: see point 8). Considering that epilepsy characterize the neurodevelopmental syndrome associated with CNANTP2 mutations, the authors need to discuss whether patient-derived organoids also show compromised neuronal function activity.

Minor concerns.

1. Fig. 1C. Explain in the legend what the two lanes in the WB represent (are they biological replicates)? The levels of actin in the WB are not homogenous: include WB on additional housekeeping genes or other loading controls. Add molecular weight corresponding to the detected protein.

2. Supplementary figure 1 is not clear. Explain in the figure legend what each dot represents (why 76 dots for 37 patients?)

3. Supplementary figure 1B: Increase the number of samples (n=5 patients and n=4 controls) to support the finding about enhanced gray matter thickness, due to the substantial intrinsic individual variability. Were the selected patients and controls male or female, what age?

4. Fig. S1D and S1C are inverted in the manuscript text (line 358: Fig. S1C should be replaced with Fig. S1D and line 360: Fig. S1D should be replaced with Fig. S1C).

5. The authors compare the results obtained with the 2D brightfield microscopy to those obtained with light sheet imaging to confirm that the 2D brightfield images can be used for reliably quantifying organoid volume. Indeed they both show a 1.5-fold increase in area of patient- versus control- derived organoids. However, for the bright-field imaging they use 13 week organoids, while for light-sheet they use 26 weeks organoids. In order to compare the results obtained by using these two imaging approaches, the authors have to calculate and compare the fold increase of the same organoids or at least of organoids with the same age.

6. line 418. Fig. S2A should be replaced with Fig. 2E

7. Fig. 4B. Add also a control to the WB, to show whether similar or reduced protein levels are present in the rescued line, as compared to the control. Also discuss if both the alleles of CNTNAP2 have been restored, resulting in a complete or partial rescue.

8. Fig. S1J: add heat map color scale.

9. Figure S2A is not cited in the text.

Reviewer #1 (Remarks to the Author):

In the manuscript entitled “Cortical Overgrowth in a Preclinical Forebrain Organoid Model of CNTNAP2-Associated Autism Spectrum Disorder”, Jong et al. report a patient-derived organoid model of CNTNAP2 mutation. The authors generate forebrain organoids from patients carrying a homozygous c.3709delG mutation in CNTNAP2, and show that the organoids exhibit an increased volume that is consistent with the increase in gray matter volume observed in patients. They determine that the increased volume stems from the increased proliferation of neural progenitor cells, and show that this phenotype is rescued upon correction of the mutation by CRISPR-mediated gene editing. In addition, the authors show that gene expression is dysregulated in the patient-derived organoids, and determine that the differentially expressed genes include known ASD-associated genes and are enriched for genes in ASD-associated weighted gene co-expression networks.

The data is interesting, and has clinical relevance. However, the authors should consider characterizing their organoid model in more detail. There are a few points that should be addressed to strengthen the study:

1. The authors immunoblot for CNTNAP2 in 0-10 week-old organoids (Fig. 1C), and suggest that CNTNAP2 expression in their organoids is in concordance with in vivo expression patterns, making their organoids a good preclinical neuronal model system (Lines 398-400). To make a stronger argument for this, the authors should directly compare their organoids to the Brainspan data to establish: (1) What brain region(s); (2) What fetal age their organoids most closely correspond to. Given that CNTNAP2 expression varies across brain regions and developmental timepoints (Fig. S1J), it is important to confirm that the 6-13 week organoids match the brain regions and developmental timepoints of interest, making them an appropriate disease model. This comparison could be done by performing correlation analysis on the transcriptomes of the organoids and the Brainspan samples used to generate Fig. S1J. Please also include a scale for the heatmap colors in Fig. S1J.

We thank the reviewer for this suggestion. In order to address this issue, we have used our RNA bulk-seq data from the 8-week timepoint to do a bioinformatics analysis by comparing these data to the Brainspan dataset. We calculated Pearson's r for all available Brainspan samples and found the highest correlations between the transcriptomes of the organoids with multiple forebrain regions from 9-week-old human fetal brain tissue. Therefore, we further establish the face validity of the organoid model for the *CNTNAP2*-associated neurodevelopmental disorder on the basis of the maximally correlated fetal age and brain region. We have now added the following to the Results section of the revised manuscript:

Second, at the transcriptional level, we performed bulk-RNA sequencing to reveal that at 8 weeks in culture, the organoids display the highest correlations with multiple

forebrain structures from 9 week-old human fetal brain tissue, when compared to transcriptional profiles from samples from all brain structures and timepoints from the Brainspan transcriptome database (28) (Fig. 1C).

As suggested, we have also added a scale to the heatmap displaying human *CNTNAP2* expression.

2. The authors should note if there were any electrophysiological differences when they performed whole-cell patch-clamp recordings on the 8 week patient- and control-derived organoids (Lines 370-373; Figs S1G-H), given that altered network activity has been reported in the brains of *Cntnap2* knockout mice (e.g. Lazaro et al., Cell 2019) as well as patient-derived *CNTNAP2* mutant neurons. If differences are observed in patient- and control-derived organoids, the authors should then record from the CRISPR-rescued organoids to ensure that any phenotype is restored to the wildtype condition.

We thank the author for this suggestion and would like to elaborate on this issue. The purpose of our electrophysiology experiments was solely intended to show that our protocol generated neurons that do indeed reach electrophysiological maturity. Although we agree it would be very interesting to use our modeling system to investigate electrophysiological differences between patient- and control-derived organoids, such experiments are beyond the scope of this paper and warrants a separate follow-up study. In this first *CNTNAP2* forebrain organoid paper, we intended to specifically focus on early developmental processes involved in *CNTNAP2*-associated ASD.

3. Along these lines, did the authors observe any additional phenotypes in their patient-derived organoids beyond the increase in total volume? *CNTNAP2* mutations have been reported to be associated with a myriad of different phenotypes beyond macrocephaly (e.g., increased astrogliosis, delayed NPC migration), and it would be good for the authors to discuss if additional phenotypes are observed in their organoids beyond an increase in total volume. The authors do report observing altered gene expression in their patient organoids, but they should elaborate further on this (see point 5 below).

While we were not able to identify additional phenotypes previously described in the post-surgical brain tissue of patients with CDFE, and a further examination of the bulk- and single cell RNAseq data did not reveal differences in ion-channel expression (see point 5 below), we did identify additional disease phenotypes at the molecular level. The top 100 signature genes of each of *CNTNAP2*-positive cell types in control organoids were significantly associated with multiple human disease phenotypes related to CDFE, including language delay, Pitt-Hopkins syndrome, epilepsy and corpus callosum hypoplasia. Moreover, we were subsequently able to provide independently supporting evidence to show that the corpus callosum hypoplasia phenotype in brain MRIs from CDFE-patients, which has now been included in the Results section of the revised manuscript:

Disease phenotype GO analysis (49) on the cell type-specific signature genes for each of the three CNTNAP2-expressing cell types are associated with ASD, epileptic encephalopathy, Pitt-Hopkins syndrome and hypoplasia of the corpus callosum (Fig. S3N-P). The first three disease phenotypes have all been described in patients with CNTNAP2 LoF mutations (10, 50) while corpus callosum volumetric changes had not been investigated until now. In order to investigate the latter, we carried out a qualitative comparison of corpus callosum volumetric differences on MRI between the CNTNAP2 mutation carriers and matched healthy controls without the mutation and found a decrease in corpus callosum volume in patients utilizing a one-sample t-test was performed between the size deviation for the templates (which are 0), and found that mutation carriers indeed demonstrate a decrease in corpus callosum volume (t-test; $p=0.0127$) (Table S12).

4. The authors should examine their RNA-seq data for the presence of CNTNAP2 mRNA transcripts. If no transcripts are detected in the patient organoids, this could suggest that the mutation destabilizes the mRNA by inducing nonsense mediated decay, thus accounting for the lack of detectable protein (Line 388, Fig. 1D/S1I).

We thank the reviewer for this suggestion. We have now added Fig. S1X, which displays the scRNAseq data of *CNTNAP2* transcript sequencing reads. The reduced coverage of *CNTNAP2* mRNA after the mutation site is consistent with partial nonsense mediated decay of *CNTNAP2* mRNA occurring predominantly at the 3'-terminal end of the transcript. We have now modified the Results section as written:

Analyzing the mRNA sequence reads for CNTNAP2 scRNAseq confirms that the mRNA is indeed present in lower numbers, with case organoids showing a lower read-coverage at the 3' end than controls (Fig. S1K).

5. Along these lines, were there any differences observed in the expression of Kv1.1 and Nav1.2 channels in the RNA-seq data from the patient organoids? Altered expression of ion channels has been previously reported in the brain samples of human patients. It would be good to assess if this difference is also observed in the patient-derived organoids, and if it can be corrected in the CRISPR-rescued organoids.

Indeed, Strauss *et al.*, 2006, demonstrated abnormal protein expression using antibodies directed against sodium and potassium channels in post-surgical brain tissue from Amish children with ASD who carry the *CNTNAP2* c.3709DelG mutation. Moreover, the Peles lab and others have demonstrated that *CNTNAP2* plays a role in the clustering of potassium channels in the nodes of Ranvier in the peripheral nervous system.

We have examined our bulk RNAseq and scRNAseq expression data to investigate whether such a difference exists at the transcriptome level. However, we do not observe significant differential gene-expression for any type of sodium or potassium channel in

either datasets (see Table S6 and Table S10). A plausible explanation for this discrepancy could be that the phenotype of aberrant potassium and sodium channel expression only arises during a later developmental stage (e.g., postnatally) than modeled by our organoids, which reflect first and second trimester brain development. Alternatively, the changes observed by Strauss *et al.*, 2006 may have resulted from post-translational mechanisms.

6. In Fig. 3F, the 13-week patient organoids show a significantly lower fraction of PAX6+ cells as compared to control organoids ($p = 0.0008$). However, in Fig. 4H, the 13-week control and patient organoids do not appear to be significantly different in their respective fractions of PAX6+ cells (blue dots vs gray dots). Could the authors comment on this discrepancy?

We thank the reviewer for this comment. As we have added single cell transcriptome data for the 13-week time point to identify cell types based on the use of multiple markers instead of the single marker approach, we have omitted the figures referred to in the comment and replaced them with scRNAseq data.

7. The authors should specify if the CRISPR correction occurred on both alleles. If possible, the locus should be deep sequenced (e.g., amplicon sequencing) to ensure that no mutant allele is present. In addition, it would be good to assess what the level of rescue is relative to wildtype control. This could be done by updating the immunoblot in Fig. 4B to show the amount of CNTNAP2 protein in control, patient-2 and patient-2 rescue organoids.

We appreciate the opportunity to clarify this issue. Sanger sequencing was performed on PCR amplicons surrounding the mutation site in *CNTNAP2* after selecting single cell iPSC-clones that were CRISPR edited. As shown in Figure 4A, there are no double peaks at the mutation site (indicated by the red arrow), as would be expected had a heterozygous edit occurred. Therefore, we conclude that we did in fact generate a homozygous wildtype cell line for *CNTNAP2*.

We now describe this more explicitly in the Results section of the revised manuscript: *Sanger sequencing of a CRISPR-edited single-cell derived clone confirmed the homozygous wildtype DNA sequence.*

8. To improve the flow of the manuscript, the authors should consider discussing the RNA-seq data before CRISPR rescue (i.e., switching Figs. 4 and 5 around). In this way, they could completely characterize their organoids before discussing rescue by gene editing. They should also consider performing RNA-seq in their rescued organoids to determine if the altered gene expression patterns are restored to wildtype levels.

We thank the reviewer for these suggestions. In the revised version of the manuscript, we have now switched Figures 4 and 5 and the corresponding paragraphs in the Results section as suggested to improve the flow and narrative of the manuscript.

In addition, we also performed RNAseq on the CRISPR-rescue line and its parental patient-derived line as suggested by the reviewer. The data from this analysis shows that of 339 genes that were differentially expressed between case- and control-derived organoids, 67 are expressed in the opposite direction at a statistically significant level in the CRISPR-rescue line compared to its parental patient-derived line. This indicates a partial rescue of the mRNA expression levels (see figure S4B). In addition, while not reaching statistical significance, the log₂fold change also demonstrates a normalization of the expression levels of these 339 DEgenes (shown in the heatmap in figure 5G). As can be seen in the PCA plot (figure S4F), there is a considerable amount of variability in the mRNA levels between the batch containing the three case-control pairs and the batch containing the CRISPR line and its parental patient line that was performed at a later time. However, the PCA plot clearly demonstrates that the genotypes separate in the same manner on the Y-axis of the PCA plot, despite the largest separation of the data being between the two batches (figure S4E). Furthermore, the top-100 genes contributing to PC2 are significantly enriched for the initial 339 DEgenes, whereas the the top-100 genes contributing to PC1 were not enriched for these 339 DEgenes (Fig. S4F-G).

We have now added these data to the Results section:

At the bulk-transcriptional level, 576 genes were differentially expressed between organoids derived from the CRISPR-rescue line when compared to its parental patient-derived line. Of the 339 genes that were differentially expressed in patient-organoids, 67 of these were expressed in the opposite direction in the CRISPR-rescue line, indicating a partial rescue (Fig. S4B-C) (permutation analysis, $n = 10000$, $z = 18.6$, $p = 2.3e-77$). These 67 genes are enriched for biological processes related to neurodevelopment including cellular proliferation (Fig. S4D). Regardless of whether reaching the threshold for statistical significance, the gene expression directionality of the 339 DEgenes was generally reversed between patients- versus control lines and CRISPR-rescue versus patients line (Fig. 5G). PCA revealed a large batch effect in sample clustering represented by PC1 and a genotype-rescue effect represented by PC2 (Patient vs. CRISPR-rescue) (Fig. S4E and S4F). Of the 100 genes with the highest contribution to PC1 (86% of the variance) only 3 genes are present among the 339 DEgenes from the initial patient-control batch (permutation analysis; $n = 10,000$; $z = 0.99$; $p = 0.32$), whereas the top-100 contributing genes for PC2 (6% of the variance) include 41 of these 339 DEgenes (permutation analysis; $n = 10,000$; $z = 29.7$; $p = 8.86e-195$), indicating the batch effect manifests through genes different from the 339 DEgenes from the initial patient-control batch (Fig. S4G).

9. Please review the numbering of all figures. There are a number of inconsistencies, for instance Figs. S1F-H are missing, Fig. S2 does not correspond to what is listed in the text (Line 418), and Fig. 4H is not listed in the legend.

In the revised version of the manuscript, we have omitted some of the old figures and replaced it with figures that were derived from the new scRNAseq data analysis that the reviewers had asked us to carry out to address some of their concerns. We have reviewed the numbering of all the reformatted figures and verified how all figures are being referenced throughout the manuscript.

Reviewer #2 (Remarks to the Author):

This paper describes the phenotype of syndromic CNTNAP2 mutations--which are associated with marked phenotypic abnormalities--on the growth of iPSC-derived organoids. Notably, these organoids display marked volume increases, initial progenitor hyper-proliferation, and increased neurogenesis among other abnormalities. Importantly, knockout mice don't recapitulate these phenotypes nor do monolayer cultures.

The authors have written a very clear manuscript and the experiments are well designed. However, despite the remarkable phenotype, most of the experiments are "by the book" and little mechanistic investigation is done despite the nice CRISPR rescue. For example, some type of clonal labeling could clarify the natural history of cell expansion between controls and patients. Or the critical mediators downstream of the mutation could have been explored functionally. Also of note is the relative rarity of this particular gene mutation in ASD. The potential limitations of organoids should be discussed in terms of later synaptic events, compensatory events, and other things that can't be observed in such "young" organoids (compared with a neonatal or toddler brain).

We thank the reviewer for these suggestions. While performing clonal labeling to clarify the natural history of cell expansion and examining critical mediators downstream from the mutation are both areas of interest for us, we feel that they are beyond the scope of our current manuscript. Regarding the relative rarity of this particular gene mutation as a cause of ASD, we would like to point out that high-confidence ASD- genes are individually rare but jointly account for approximately 15% of ASD cases. Finally, the fact that we see an enrichment for SFARI ASD genes in both our bulk RNA sequencing and single-cell RNA sequencing experiments confirms that the *CNTNAP2*-associated pathophysiology has overlap with other forms of ASD at the molecular level.

Reviewer #3 (Remarks to the Author):

Manuscript # NCOMMS-19-26994-T

Cortical Overgrowth in a Preclinical Forebrain Organoid Model of CNTNAP2-Associated Autism Spectrum Disorder.

In this study, de Jong et al. investigate effects of homozygous c.3709DelG

mutation of CNTNAP2 by using forebrain organoids generated from iPSC derived from patients from the Old Order Amish community carrying this mutation, that present with an increased head circumference and increased gray matter volume. The authors observed a similar volume increase in patient-derived organoids, that they linked to an enhanced proliferation of neuronal progenitors, that leads to an increase in the generation of cortical neurons and non-neuronal cell types. Remarkably, the observed phenotypes were rescued by correcting the CNTNAP2 gene mutation.

This work aims to demonstrate that organoids can be used to understand critical roles of genes associated with ASD. However, the overall conclusions are overstated and not supported by appropriate scientific evidence. Additional analysis needs to be performed to confirm these findings.

Major concerns.

1. The protocol for organoid generation is based on the method previously described by Kadoshima et al. 2013, however it introduces some modifications, e.g. embryoid bodies are cultured for 5 days in mTESR1 before neural differentiation is induced (through TGF β and WNT inhibition). The effect of such modifications on the cell types generated by using the originally described protocol are not known. A detailed analysis (e.g. by using immunohistochemistry) for different cortical markers across time points, needs to be included to help understand how the cellular composition changes over time and to support the choice of using organoids cultured for either 6 or 13 weeks, to underline cell-type phenotypes associated with disease.

We thank the reviewer for this comment and agree that the modifications in the organoid generation protocol we used had not been tested adequately to confirm the generation of the desired forebrain organoids. We have therefore conducted two additional experiments to confirm that we did indeed generate forebrain organoids displaying the desired characteristics at the two specific time points of investigation: 1) We have performed single-cell RNA sequencing at 13 weeks *in vitro* and show that the cell types generated at this time point correspond with cell types present in human embryonic fetal brain as has been described by Nowakowski *et al.* (2017). These include astrocytes, oligodendrocyte precursor cells, microglia, radial glia, intermediate progenitor cells, excitatory cortical neurons, ventral MGE progenitors, inhibitory cortical interneurons, choroid plexus cells, mural cells, and endothelial cells. 2) We have added immunohistochemistry data showing expression of early born cortical neuron markers *TBR1* and *CTIP2*. These data are in addition to the IHC data of *PAX6* at 6 weeks *in vitro* and *FOXP1* and *MAP2* at 4 weeks *in vitro* provided in the previous version of the manuscript.

We have now modified the Results section of the revised manuscript as follows:

At 13 weeks in vitro, organoids displayed expression of early born cortical layer marker TBR1 and CTIP2 as shown using immunostaining (Fig. S1G).

Third, we evaluated whether the cell type composition of the organoids resembles that of early human fetal prefrontal brain, by performing scRNAseq on 13 week-old organoids. Unsupervised Leiden clustering revealed 26 distinct clusters that were consistently present in all case- and control-derived organoids (Fig. S1H and table S4). We annotated these cell populations using PFC cell type signatures previously identified by scRNAseq of human embryonic PFC tissue (average age: 16.3 post conceptional weeks (pcw)) (36). These analyses demonstrate that the forebrain organoids contain a range embryonic prefrontal cortical cell types (Fig. 1D; Fig. S1I and table S5).

2. One of the big finding of the manuscript is that patient- and control- derived forebrain organoids can recapitulate differences observed in patients, such as increased volumes. How the increased volume ratios change over time needs to be investigated further. At least, include analysis of volumes at week 6, that is when the highest increase in cell number is observed. IHC analysis at the same time points also have to be included (see point1).

We thank the reviewer for this suggestion. In the revised version of the manuscript, we now display the growth trajectory of case- and control-derived organoids at 3 different time points: 4-, 9- and 13 weeks *in vitro*. Figure 2B has been replaced by a figure containing the 9-week time point.

We have now added the following text to the Results section:

After 4 weeks in culture, case- and control organoids that were generated from an equal number of hiPSCs display no difference in measured surface area. At 9 weeks, patient-derived organoids increase in surface area relative to control-derived organoids, and at 13 weeks in vitro, patient-derived organoids showed a 1.5-fold increase in surface area compared to control-derived organoids (LRT, $p = 0.002$, $n = 6-7$ organoids/line) (Fig 2B).

3. It has been previously shown that the organoid models are plagued by poor reproducibility (Quadrato et al. Nature 2017). The authors do not show or discuss the degree of reproducibility of cell type generation by using their organoid model. Unless data showing high levels of reproducibility of cell types generated by individual organoids are provided, a larger number of organoids need to be analyzed to support the findings.

We thank the reviewer for bringing up this issue and agree that one of the major challenges in the organoid field is the considerable degree of heterogeneity that exists on multiple levels – even between organoids grown from the same cell line from the same batch of organoids. For this reason, we have utilized our newly obtained scRNAseq data to show that, although there is the expected amount of variability in the fractions of cell types that are being generated, all clusters as defined by PCA are indeed present in all cell lines (Fig S1I).

We have now added these findings to the Results section of the revised manuscript:

Unsupervised Leiden clustering revealed 26 distinct clusters that were consistently present in all case- and control-derived organoids (Fig. S1H and table S4).

4. One of the major findings of the manuscript is that patient-derived organoids show an increase in volume. However, the images in figure 3C show actually the opposite (mutant organoids are smaller than control). Are the images representatives of the sample?

We thank the reviewer for this comment. The section in the control panel of figure 3C was a section that was taken from a region closer to the edge of the organoid, whereas the section in the case panel was taken from a section of the case-derived organoids closer to its center. Therefore, the control-derived organoid section had a smaller surface area than the section of the case-derived organoid. In the revised version of the manuscript, we have now added images that are more representative of the sample.

5. Conclusions about progenitor proliferation and premature neurogenesis are exclusively based on immunohistochemistry data obtained by using a limited number of markers (PAX6, Ki67 and NeuN). However, the assessment of single markers is not sufficient to define cell types. Single cell RNA-seq analysis would be the preferred method to define specific gene signatures that identify cell types and importantly, to claim that those are affected by the CNTNAP2 mutation. However, if this is not possible, at least analysis of a larger number of markers, to assess different cell types and biological processes, has to be included to support the conclusions the authors make.

We appreciate the opportunity to address this fundamental issue and fully agree with the reviewer that single-marker analyses are not the ideal way to reliably identify cell types. As suggested, we have now performed scRNAseq to establish cell type properties, thereby extending the earlier single-marker neuronal cell type quantifications. We have also used these scRNAseq data to determine which cell types express *CNTNAP2* and are thus affected by the mutation.

We have now added the following text to the Results section of the revised manuscript:

Third, we evaluated whether the cell type composition of the organoids resembles that of early human fetal prefrontal brain, by performing scRNAseq on 13 week-old organoids. Unsupervised Leiden clustering revealed 26 distinct clusters that were consistently present in all case- and control-derived organoids (Fig. S1H and table S4). We annotated these cell populations using PFC cell-type gene expression signatures previously identified by scRNAseq of human embryonic PFC tissue (average age: 16.3 post conceptional weeks (pcw)) (36). These analyses demonstrate that the forebrain organoids contain a range embryonic prefrontal cortical cell types (Fig. 1D; Fig. S1I and table S5).

6. The authors claim that forebrain organoids serve as preclinical neuronal model

system to study the effects of homozygous LoF mutations on early human cortical development, based on the observation that the CNTNAP2 expression pattern in organoids is in concordance with in vivo gene expression. However, this concordance is based on comparison between protein levels assessed by immunoblotting (organoids) and gene expression assessed by bulk transcriptome analyses (human samples). If the authors want to make this claim, they need to investigate the similarity of cell types in their organoid system compared to that found in the developing human brain, at single cell level, as done in previous works (Camp et al. 2015). Otherwise, the authors need to resize this statement.

We thank the reviewer for this comment, and agree that having single-cell transcriptome data comparing the similarity of cell types in the organoid system to the human developing brain would allow us to make this claim more convincingly. We therefore have analyzed the scRNAseq data and annotated the defined clusters to cell types as defined by Nowakowski and colleagues (2017). In addition, we show that *CNTNAP2*-expressing cell types in our organoid model system are predominantly PFC excitatory neurons and that *CNTNAP2* expression is markedly reduced in other cell types (Fig 1G). This is consistent with scRNAseq data from human embryonic PFC tissue (Nowakowski et al science 2017), in which *CNTNAP2* expression is also highest in neuronal cell types (Fig. S1N).

We have now added the following text to the Results section of the revised manuscript:

Analysis of scRNAseq data indicates that at 13 weeks in vitro, in control-organoids, CNTNAP2 is most highly expressed in different types of PFC-excitatory neurons as annotated previously (36): ‘Early Born Deep Layer/subplate Excitatory Neuron PFC’ (EN-PFC), ‘Early Born Deep Layer/subplate Excitatory Neuron V1’ (EN-V1) and ‘New born excitatory neurons’ (nEN) (hereafter referred to as ‘CNTNAP2-expressing cell types’). In patient derived-organoids, CNTNAP2-expression is markedly lower in these cell types (Fig. 1G). This is in line with cell type expression pattern in human PFC cell types, with CNTNAP2 being expressed most highly in excitatory and inhibitory neuronal cell types (Fig. S1M).

7. Previous work showed that Cortical Dysplasia Focal Epilepsy (CDFE) is more common in males (Ortiz-González et al., 2013). Instead, all the data presented in the current study are generated by using organoids derived from female patients iPSc. Is there any reason why the authors decided to focus on cell from female patients? Do they expect to find similar results in organoids derived from male patient iPSCs?

We thank the author for posing this question. The manuscript by Ortiz-Gonzales and colleagues referred to here describes a difference found in the prevalence in focal cortical dysplasia between boys and girls, without establishing which genetic etiology that is driving the phenotype in these patients.

For Cortical Dysplasia Focal Epilepsy *Syndrome*, which has been specifically linked to the c.3709DelG mutation in *CNTNAP2* (Strauss et al., 2006) and which has been the focus of this study, there is no evidence for an asymmetric distribution between males and females. Given the fact that CDFE-syndrome is a Mendelian disorder segregating in an autosomal recessive manner, we have no reason to believe CDFE syndrome is more common in males than in females. This was confirmed by personal communication with Dr. Kevin Strauss, who is involved in providing clinical care of over 70 patients with the c.3709DelG mutation.

We generated cell lines from 3 female patients, as these participants were available and eligible for the study and relatively close in age at time of blood sample collection. We then recruited 3 sex-matched controls from the Old Order Amish community.

8. Fig. S1F, G and H are missing.

We appreciate the reviewer's careful reading. This has been corrected in the revised version of the manuscript.

9. The authors claim that organoids from both patients and controls show features of mature neurons (data from Fig. S1F-H, that are missing in the manuscript: see point 8). Considering that epilepsy characterize the neurodevelopmental syndrome associated with *CNTNAP2* mutations, the authors need to discuss whether patient-derived organoids also show compromised neuronal function activity.

The purpose of our electrophysiology experiments was solely intended to show that our protocol generated neurons that do indeed reach electrophysiological maturity. Although we agree it would be very interesting to use our modeling system to investigate electrophysiological differences between patient- and control-derived organoids, such experiments are beyond the scope of this paper and warrants a separate follow-up study. In this first *CNTNAP2* forebrain organoid paper, we intended to specifically focus on early developmental processes involved in *CNTNAP2*-associated ASD.

Minor concerns.

1. Fig. 1C. Explain in the legend what the two lanes in the WB represent (are they biological replicates)? The levels of actin in the WB are not homogenous: include WB on additional housekeeping genes or other loading controls. Add molecular weight corresponding to the detected protein.

We thank the reviewer for this comment. The two lanes per sample are technical replicates. In each lane, we have loaded an equal amount of protein. There is indeed some variation in the relative amount of actin in each sample, with the amount of actin increasing with time relative to the total amount of protein, which is expected in the context of the maturing organoids. We have quantified the amount of *CNTNAP2* protein

relative to the amount of actin, so the initial lower amount of actin should not affect the quantification.

For clarification, we have edited the legend for figure 2C:

Western blot analysis showing increasing CNTNAP2 expression levels over time relative to actin in a control-derived organoid line, roughly corresponding to the developmental pattern of CNTNAP2 expression pattern in human embryonic brain. Each sample contains two lanes containing technical replicates; the bar graph represents the mean quantification from these two lanes.

2. Supplementary figure 1 is not clear. Explain in the figure legend what each dot represents (why 76 dots for 37 patients?).

We thank the reviewer for pointing out this issue. In total, there were 76 measurements among the 37 individual patients.

We have now clarified this in the figure legend:

Plotted are 76 longitudinal head circumference measurements from 20 unique female and 17 unique male pediatric patients between birth and 5 years from the Old Order Amish community, who were diagnosed with ASD and carried the c.3709DelG mutation in CNTNAP2. Z-scores were calculated per month and gender using mean and SD from WHO-reference data (weighted z-score = 2.93, $p < 0.003$).

3. Supplementary figure 1B: Increase the number of samples (n=5 patients and n=4 controls) to support the finding about enhanced gray matter thickness, due to the substantial intrinsic individual variability. Were the selected patients and controls male or female, what age?

We agree with the reviewer that the sample size of the MRI analysis is small. However, we were unable to obtain more MRIs from patients carrying the *CNTNAP2* c.3709DelG mutation. As this is a highly rare mutation that has only been reported among the Old Order Amish, MRI scans from these patients are not easily obtained. That being said, all 5 subjects are from a genetic isolate/founder population with a considerable degree of genetic homogeneity, thereby providing a more robust background compared to studies involving unrelated subjects. In the revised version of the manuscript, we have added a table that contains information related to the sex and age of the subjects from whom brain MRI data is available.

4. Fig. S1D and S1C are inverted in the manuscript text (line 358: Fig. S1C should be replaced with Fig. S1D and line 360: Fig. S1D should be replaced with Fig. S1C).

We thank the reviewer for this comment. In the revised version of the manuscript, we have corrected this as suggested.

5. The authors compare the results obtained with the 2D brightfield microscopy to those obtained with light sheet imaging to confirm that the 2D brightfield images can be used for reliably quantifying organoid volume. Indeed they both show a 1.5-fold increase in area of patient- versus control- derived organoids. However, for the bright-field imaging they use 13 week organoids, while for light-sheet they use 26 weeks organoids. In order to compare the results obtained by using these two imaging approaches, the authors have to calculate and compare the fold increase of the same organoids or at least of organoids with the same age.

We thank the reviewer for this suggestion. Unfortunately, due to practical issues related to lab accessibility and microscope availability during the coronavirus pandemic, we were not able to generate 3D light-sheet data at the desired time-point. Instead, we now display the growth trajectory of case- and control-derived organoids at 3 different time points: 4-, 9- and 13 weeks *in vitro* using the 2D approach to validate the ontology of the volumetric increase.

6. Fig. S2A should be replaced with Fig. 2E

We have corrected this in the revised version of the manuscript.

7. Fig. 4B. Add also a control to the WB, to show whether similar or reduced protein levels are present in the rescued line, as compared to the control. Also discuss if both the alleles of CNTNAP2 have been restored, resulting in a complete or partial rescue.

We have performed an additional Western blot analysis comparing the CNTNAP2 protein levels of the CRISPR rescue line, its parental mutant line and two control lines. These data confirm that CNTNAP2 protein levels of the CRISPR rescue line have been restored to wildtype levels.

This finding has now been updated in the Results section of the revised manuscript:

Western blotting confirmed the presence of the CNTNAP2 protein in the organoids derived from the CRISPR-line, at levels similar to those of control-derived organoids (Fig. 5B)

8. Fig. S1J: add heat map color scale.

We have added the heat map color scale to this figure in the revised version of the manuscript.

9. Figure S2A is not cited in the text.

We have corrected this in the revised version of the manuscript.

Reviewers' comments:

Reviewer #1 (Remarks to the Author):

My concerns have been largely addressed by the revision experiments that the authors have done.

Reviewer #2 (Remarks to the Author):

In general, I approve of the revised manuscript. The single-cell sequencing is helpful and has the potential to clarify a significant amount of issues.

In terms of the developmental phenotype--which is the key feature--the manuscript continues to be superficial in this regard. There are any number of ways this could be readily addressed but the authors seem unwilling.

Reviewer #3 (Remarks to the Author):

In the revised manuscript the authors have included a substantial amount of additional data to directly address my concerns. I would like to thank the authors for the effort and time they invested to address my comments. I think that the manuscript has now substantially improved.

I appreciate that the authors included immunohistochemistry and scRNA-seq data to characterize further the cell types and features of the forebrain organoid model they used in this study. The comparison of gene signatures to pre-existing datasets of endogenous cell types (Nowakowski et al. 2017) is valuable, especially to identify what fetal age the organoids most closely resemble. However, using an automatic cell type assignment method trained on the transcriptional profiles from the human fetal brain might have some limitations. In my opinion, it is very unlikely that the forebrain organoid model used in this study contains endothelial and microglia cells, that have a different embryonic origin (yolk sac). I also doubt that the interneurons in this model are MGE-like, considering that the protocol used doesn't include any ventral patterning signal (e.g. SHH agonists), and more likely represents a dorsal forebrain organoid model.

Therefore, I suggest to revise the cell type assignments to include a more accurate labeling of the different cell types contained in figure 1D, 3H and S1H. Adding a supplementary figure with the tSNE plots for the specific genes expressed by the different cell types would also be good.

REVIEWERS' COMMENTS

Reviewer #1 (Remarks to the Author):

My concerns have been largely addressed by the revision experiments that the authors have done.

We would like to thank the reviewer for his positive feedback that our revision addressed his concerns.

Reviewer #2 (Remarks to the Author):

In general, I approve of the revised manuscript. The single-cell sequencing is helpful and has the potential to clarify a significant amount of issues.

In terms of the developmental phenotype--which is the key feature--the manuscript continues to be superficial in this regard. There are any number of ways this could be readily addressed but the authors seem unwilling.

We would like to thank the reviewer for approving on our revised manuscript. We will gain much more developmental details in our future project.

Reviewer #3 (Remarks to the Author):

In the revised manuscript the authors have included a substantial amount of additional data to directly address my concerns. I would like to thank the authors for the effort and time they invested to address my comments. I think that the manuscript has now substantially improved.

I appreciate that the authors included immunohistochemistry and scRNA-seq data to characterize further the cell types and features of the forebrain organoid model they used in this study. The comparison of gene signatures to pre-existing datasets of endogenous cell types (Nowakowski et al. 2017) is valuable, especially to identify what fetal age the organoids most closely resemble. However, using an automatic cell type assignment method trained on the transcriptional profiles from the human fetal brain might have some limitations. In my opinion, it is very unlikely that the forebrain organoid model used in this study contains endothelial and microglia cells, that have a different embryonic origin (yolk sac). I also doubt that the interneurons in this model are MGE-like, considering that the protocol used doesn't include any ventral patterning signal (e.g. SHH agonists), and more likely represents a dorsal forebrain organoid model.

Therefore, I suggest to revise the cell type assignments to include a more accurate labeling of the different cell types contained in figure 1D, 3H and S1H. Adding a supplementary figure with the tSNE plots for the specific genes expressed by the different cell types would also be good.

We would like to thank the reviewer for his positive feedback on our revision that addressed his concerns and substantially improved manuscript. We thank the reviewer for his additional suggestions to improve our paper. We agree that the classification of the presence of microglia and endothelial cells is biologically unlikely and likely due to misclassification in automatic annotation approaches. This misclassification has been shown by other previously studies published (<https://www.ncbi.nlm.nih.gov/pmc/articles/PMC7433012/>). We revised the methods in the main text "Although our correlation annotation approach annotated a small set of endothelial and microglia cells, our further analysis using predefined biomarkers (adapted from <https://www.ncbi.nlm.nih.gov/pmc/articles/PMC7433012/> Suppl Table2) indicated that the expression of key biomarkers of these cell types are either absent or weak as compared to other neural types (Suppl Figure 1J). Therefore, we omitted these cell types for further downstream analysis." We removed the cell annotation from Figures 1D and S1I. In addition, we added a supplementary dot plot (Suppl Figure 1J) to show the expression pattern of the predefined cell type markers as a confirmation.

For the interneuron annotation, existence of interneurons and their progenitors in the forebrain organoids have been specifically described by Velasco et al (<https://www.nature.com/articles/s41586-019-1289-x>). They used a similar differentiation approach as we did, and they noted that the origin of these interneurons is unclear. Therefore, we believe that the interneurons detected in our organoids are due to the complexity of 3D organoid culture instead of the artifacts in cell type annotation. We prefer to keep the annotation of the interneurons and their progenitors.